

# Automated black-box boundary value detection

Felix Dobslaw[1], Robert Feldt[2] and Francisco Gomes de Oliveira Neto[3]

[1] Department of Communication, Quality Management and Information Systems, Mid Sweden University, Östersund, Jämtland, Sweden

[2] Department of Computer Science and Engineering, Chalmers University of Technology, Gothenburg, Västra Götaland, Sweden

[3] Department of Computer Science and Engineering, University of Gothenburg, Gothenburg, Västra Götaland, Sweden

## ABSTRACT

Software systems typically have an input domain that can be subdivided into subdomains, each of which generates similar or related outputs. Testing it on the boundaries between these sub-domains is critical to ensure high-quality software. Therefore, boundary value analysis and testing have been a fundamental part of the software testing toolbox for a long time and are typically taught early to software engineering students. Despite its many argued benefits, boundary value analysis for a given software specification or application is typically described in abstract terms. This allows for variation in how testers apply it and in the benefits they see. Additionally, its adoption has been limited since it requires a specification or model to be analysed. We propose an automated black-box boundary value detection method to support software testers in performing systematic boundary value analysis. This dynamic method can be utilized even without a specification or model. The proposed method is based on a metric referred to as the program derivative, which quantifies the level of boundariness of test inputs. By combining this metric with search algorithms, we can identify and rank pairs of inputs as good boundary candidates, i.e., inputs that are in close proximity to each other but with outputs that are far apart. We have implemented the AutoBVA approach and evaluated it on a curated dataset of example programs. Furthermore, we have applied the approach broadly to a sample of 613 functions from the base library of the Julia programming language. The approach could identify boundary candidates that highlight diverse boundary behaviours in over 70% of investigated systems under test. The results demonstrate that even a simple variant of the program derivative, combined with broad sampling and search over the input space, can identify interesting boundary candidates for a significant portion of the functions under investigation. In conclusion, we also discuss the future extension of the approach to encompass more complex systems under test cases and datatypes.

# INTRODUCTION

Ensuring software quality is critical, and while much progress has been made to improve formal verification approaches, testing is still the *de facto* method and a crucial part

Corresponding author
Felix Dobslaw, felix.dobslaw@miun.se

of modern software development. A central problem in software testing is how to meaningfully and efficiently cover an input space that is typically very large. A fundamental, simplifying assumption is that even for such large input spaces, there are subsets of inputs, called partitions or sub-domains, that the software will handle in the same or similar way (*Goodenough & Gerhart, 1975*; *Richardson & Clarke, 1985*; *Grindal, Offutt & Andler, 2005*). Thus, if we can identify such partitions, we only need to select a few inputs from each partition to test and ensure they are correctly handled.

While many different approaches to partition testing have been proposed (*Goodenough & Gerhart, 1975*; *Richardson & Clarke, 1985*; *Ostrand & Balcer, 1988*; *Hamlet & Taylor, 1990*; *Grochtmann & Grimm, 1993*), one that comes naturally to many testers is boundary value analysis (BVA) and testing (*Myers, 1979*; *White & Cohen, 1980*; *Clarke, Hassell & Richardson, 1982*). It is based on the intuition that developers are more likely to get things wrong around the boundaries between input partitions, *i.e.,* where there should be a change in how the input is processed and in the output produced (*Clarke, Hassell & Richardson, 1982*). By analysing a specification, testers can identify partitions and boundaries between them. They should then select test inputs on either side of these boundaries and thus, ideally, verify both correct behaviours in the partitions and that the boundary between them is in the expected, correct place (*Clarke, Hassell & Richardson, 1982*; *British Computer Society, 2001*). But note that identifying the boundaries, and thus partitions, is the critical step; a tester can then decide whether to focus only on them or sample some non-boundary inputs "inside" of each partition.

Empirical studies on the effectiveness of partition and boundary value testing do not provide a clear picture. While early work claimed that random testing was as or more effective (*Hamlet & Taylor, 1990*), they were later countered by studies showing clear benefits to BVA (*Reid, 1997*; *Yin, Lebne-Dengel & Malaiya, 1997*). A more recent overview of the debate also provided theoretical results on the effectiveness and discussed the scalability of random testing in relation to partition testing methods (*Arcuri, Iqbal & Briand, 2011*). Regardless of the relative benefits of the respective techniques, we argue that improving partition testing and boundary value analysis has both practical and scientific value; judging their value will ultimately depend on how applicable and automatic they can be made.

A problem with partition testing, in general, and BVA, in particular, is that there is no clear and objective method for identifying partitions or the boundaries between them. Already *Myers (1979)* pointed out the difficulty of presenting a "cookbook" method and that testers must be creative and adapt to the software being tested. Later work describes BVA and partition testing as relying on either a partition model (*British Computer Society, 2001*), categories/classifications of inputs or environment conditions (*Ostrand & Balcer, 1988*; *Grochtmann & Grimm, 1993*), constraints (*Richardson & Clarke, 1985*; *Ostrand & Balcer, 1988*), or checkpoints (*Yin, Lebne-Dengel & Malaiya, 1997*) that are all to be derived from the specification. But they do not guide this derivation step in detail. Several authors have also pointed out that existing methods do not give enough support to testers (*Grindal, Offutt & Andler, 2005*) and that BVA is, at its core, a manual and creative process that cannot be automated (*Grochtmann & Grimm, 1993*). More recent work can be seen as

overcoming the problem by proposing equivalence partitioning and boundary-guided testing from formal models (*Hübner, Huang & Peleska, 2019*). However, this assumes that such models are(readily) available or can be derived and maintained without major costs.

One alternative is to view and provide tooling for using BVA as a white-box testing technique. *Pandita et al. (2010)* use instrumentation of control flow expressions and dynamic, symbolic execution to generate test cases that increase boundary value coverage (*Kosmatov et al., 2004*). However, it is unclear how such boundaries, internal to the code, relate to the boundaries that traditional, black-box BVA would find. Furthermore, it requires instrumentation and advanced tooling, which might be costly and unavailable. It should be noted that white-box testing is limited to situations where source code is available; black-box testing approaches do not have this limitation.

Here we address the core problem of how to automate black-box boundary value analysis. We build on our recent work that proposed a family of metrics to quantify the boundariness of software inputs (*Feldt & Dobslaw, 2019*) and combine it with search and optimization algorithms to detect promising boundary candidates automatically. These can then be (interactively) presented to testers and developers to help them explore meaningful boundary values and create corresponding test cases (*Dobslaw, de Oliveira Neto & Feldt, 2020*). Our search-based, black-box, and automated boundary value detection method does not require manual analysis of a specification nor the derivation of any intermediate models. It can even be used when no specifications nor models are available. Since it is based on generic metrics of boundariness, it can principally be applied even for software with non-numeric, structured, and complex inputs and/or outputs. However, for brevity and since the overall approach is novel, we focus on implementing our AutoBVA method on testing software with arbitrarily many numeric arguments but with any type of output. In future work, we will utilize the existing hooks and empirically evaluate AutoBVA for ever more complex software to better understand the critical parts and to understand to what extent manual intervention is required for higher-level software interfaces. In this article, our main contributions are:

- A generic method for and implementation of automated boundary value analysis (AutoBVA) that uses a simple and fast variant of the program derivative (*Feldt & Dobslaw, 2019*) for quickly searching for boundary behaviour. One of the boundaries found by AutoBVA revealed inconsistencies between the implementation and documentation in the Julia language. (https://github.com/JuliaLang/julia/pull/48973)
- The comparison of two random sampling strategies within the tool: uniform and bituniform, as well as a more sophisticated heuristic local-search algorithm, and
- An empirical evaluation of AutoBVA on four curated and 654 broadly sampled software under test (SUT) to understand its capabilities in detecting boundary candidates. The code and artefacts from our experiment are available in a replication package. (https://doi.org/10.5281/zenodo.7677012)

Our results show that the proposed method can be effective even when using a simple and fast program derivative. We also see that the choice of sampling strategy affects the efficiency of boundary candidate detection-uniform random sampling without the use of

compatible type sampling does, in most cases, perform poorly (see Appendix A). For some investigated programs, the two heuristic local search strategies complement each other in the boundary-finding capabilities.

The rest of this article is organized as follows. After providing a more detailed background and overview of related work in the 'Related Work' section, we present AutoBVA in 'Automated Boundary Value Analysis'. The empirical evaluation is detailed in 'Empirical Evaluation' followed by the results in 'Results and Analysis'. The results are discussed in 'Discussion', and the article concludes in the 'Conclusions' section. Appendix A and B contain details of two screening studies that supported AutoBVA meta-parameter choices for its detection and summarisation phases, respectively. An earlier version of this article has previously been made available as a pre-print (https://arxiv.org/abs/2207.09065); consequently several parts of this article overlap heavily with that pre-print.

## RELATED WORK

In the following, we provide a brief background to boundary value analysis and the related partition testing concepts of domain testing and equivalence partitioning.

*White & Cohen (1980)* proposed a domain testing strategy that focuses on identifying boundaries between different (sub-)domains of the input space and ensuring that boundary conditions are satisfied. As summarised by *Clarke, Hassell & Richardson (1982)*: "Domain testing exploits the often observed fact that points near the boundary of a domain are most sensitive to domain errors. The method proposes the selection of test data on and slightly off the domain boundary of each path to be tested.". This is clearly connected to the testing method typically called boundary value analysis (BVA), first described more informally by *Myers (1979)* but later also included in software testing standards (*Reid, 1997*; *Reid, 2000*; *British Computer Society, 2001*). *Jeng & Weyuker (1994)* even describe domain testing as a sophisticated version of boundary value testing.

While *White & Cohen (1980)* explicitly said their goal was to "replace the intuitive principles behind current testing procedures by a methodology based on a formal treatment of the program testing problem" this has not led to automated tools, and a BVA is typically performed manually by human testers. Worth noting is also that while boundary value analysis is typically described as a black-box method (*Myers, 1979*; *Reid, 1997*; *British Computer Society, 2001*), requiring a specification, the White and Cohen papers are less clear on this, and their domain testing strategy could also be applied based on the control flow conditions of an actual implementation.

The original domain testing paper (*White & Cohen, 1980*) made several simplifying assumptions, such as the boundary being linear, defined by "simple predicates", and that test inputs are continuous rather than discrete. While none of these limitations should be seen as fundamental, they do leave a practitioner in a difficult position since it is not explicit what the method entails when some or all of these assumptions are not fulfilled. Even though later formulations of BVA as a black-box method (*British Computer Society, 2001*) avoid these assumptions, they, more fundamentally, do not give concrete guidance to testers on how to identify boundaries or the partitions they define.

[1]An annexe to the standard does provide an example of how to find partitions and identify boundaries, but the specification used in the example explicitly states the boundaries, so the identification task is trivial.

As one example, the BCS standard (*British Computer Society, 2001*) states that "(BVA) ...uses a model of the component that partitions the input and output values of the component into ordered sets with identifiable boundaries." and that "a partition's boundaries are normally defined by the values of the boundaries between partitions, however where partitions are disjoint the minimum, and maximum values in the range which makes up the partition are used" but do not give guidance on where to find or how to create such a partition model[1] if none is already at hand. This problem was clear already from Myers's original description of BVA (*Myers, 1979*), which stated, "It is difficult to present a cookbook for boundary value analysis since it requires a degree of creativity and a certain amount of specialisation to the problem at hand".

Later efforts to formalise BVA ideas have not addressed this. For example, *Richardson & Clarke (1985)* partition analysis method makes a clear difference between partitions derived from the specification *versus* from the implementation and proposes to compare them but relies on the availability of a formal specification and does not detail how partitions can be derived from it. *Jeng & Weyuker (1994)* proposed a simplified and generalized domain testing strategy with the explicit goal of automation but only informally discussed how automation based on white-box analysis of path conditions could be done.

A very different approach is *Pandita et al. (2010)*, which presents a white-box automated testing method to increase the Boundary Value Coverage (BVC) metric (originally presented by *Kosmatov et al. (2004)*). The core idea is to instrument the SUT with additional control flow expressions to detect values on either side of existing control flow expressions. An existing test generation technique to achieve branch coverage (*Pandita et al. (2010)* uses the dynamic symbolic execution test generator Pex) can then be used to find inputs on either side of a boundary. The experimental results were encouraging in that BVC could be increased by 23% on average and also lead to increases (11% on average) in the fault-detection capability of the generated test inputs.

There have been several studies that empirically evaluate BVA. While an early empirical study by *Hamlet & Taylor (1990)* found that random testing was more effective, its results were challenged in later work (*Reid, 1997*; *Yin, Lebne-Dengel & Malaiya, 1997*). *Reid (1997)* investigated three testing techniques on a real-world software project and found that BVA was more effective at finding faults than equivalence partitioning and random testing. *Yin, Lebne-Dengel & Malaiya (1997)* compared a method based on checkpoints, manually encoding qualitatively different aspects of the input space, combined with antirandom testing[2] to different forms of random testing and found the former to be more effective. The checkpoint encoding can be seen as a manually derived model of essential properties of the input space and, thus, indirectly defines potentially overlapping partitions.

[2]A form of diversity-driven test generation that also relates clearly to what is recently more commonly referred to as adaptive random testing (ART).

Even recent work on automating partition and boundary value testing has either been based on manual analysis or required a specification/model to be available. *Hübner, Huang & Peleska (2019)* recently proposed a novel equivalence class partitioning method based on formal models expressed in SysML. An SMT solver is used to sample test inputs inside or on the border of identified equivalence classes. They compared multiple variants of their proposed technique with conventional random testing. The one that sampled 50% of test

inputs within and 50% on the boundaries between the equivalence partitions performed best as measured by mutation score.

Related work on input space modelling has also been done to improve combinatorial testing. *Borazjany et al. (2013)* proposed to divide the problem into two phases, where the first models the input structure while the latter models the input parameters. They propose that ideas from partition testing can be used for the latter stage. However, for analysing the input structure, they propose a manual process that can support only two types of input structures: flat (*e.g.*, for command line parameters that have no apparent relation) or graph-structured (*e.g.*, for XML files for which the tree structure can be exploited).

We have previously proposed a family of metrics to quantify the boundariness of pairs of software inputs (*Feldt & Dobslaw, 2019*). This generalises the classical definition of functional derivatives in mathematics, which we call program derivatives. Instead of using a standard subtraction ("-") operator to measure the distance between inputs and outputs, we leverage general, information theoretical results on quantifying distance and diversity. We have previously used such measures to increase and evaluate test diversity's benefits (*Feldt et al., 2008*; *Feldt et al., 2016*). In a recent study, we used the program derivatives to explore input spaces and visualize boundaries for testers and developers (*Dobslaw, de Oliveira Neto & Feldt, 2020*). Here, we automate this approach by coupling it to search and optimization algorithms.

In summary, early results on partition and boundary value analysis/testing require a specification and do not provide detailed advice or any automated method to find boundary values. One automated method has been proposed, but it requires a system model of the SUT. One other method can automatically increase a coverage metric for boundary values but is white-box and requires both instrumentation of the SUT as well as advanced test generation based on symbolic execution. In contrast to existing research, we propose an automated, black-box method to identify boundary candidates that is simple to implement for integer arguments and conceptually extendable to arbitrary data, even with complex structures.

## AUTOMATED BOUNDARY VALUE ANALYSIS

We propose to automate boundary value analysis by a detection method that outputs a set of input/output pairs that are then summarised and presented to testers. Figure 1 shows an overview of our proposed approach. The two main parts are *detection* (on the left), which produces a list of promising boundary candidates that are then *summarised* and presented to the tester (on the right). The boundary value *detection* method is based on two key elements: (1) *search to explore* (both globally and locally) the input domain coupled with (2) the *program derivative* to quantify the boundariness of input pairs. While exploration acts as a new boundary candidate input pair generator, the program derivative acts as a filter and selects only promising candidate pairs for further processing. An archive is updated with the new pairs for summarisation, and only the unique and *good* boundary candidates are kept. The final list of promising candidates in the archive is then summarised and presented to the tester, who can select the most interesting, meaningful, or surprising

**Figure 1   AutoBVA framework for automated boundary value analysis.**

ones to turn them into test cases or start a dialogue regarding the intent with the product owner. For the summarisation step, we propose using clustering to avoid showing multiple candidates that are very similar to each other.

In this section, we describe the three main parts of our approach: selection in 'Selection: Program Derivative', search/exploration in 'Exploration: Generation of Candidate Pairs', and summarisation in 'Summarisation: validity-value similarity clustering'. Since selection is the most critical step, we first formally define the program derivative and exemplify its application by the simple bytecount SUT in 'Example: program derivative for bytecount'. It follows an explanation of the AutoBD search/exploration with its global search component ('Global Sampling') and two alternative local search strategies 'Local Neighbour Sampling' (LNS) and 'Boundary Crossing Search' (BCS). Finally, the Summarisation section introduces our approach to categorizing boundary candidates into coarse-grained validity groups by taking advantage of output types to then cluster them individually.

## Selection: program derivative

We argue that the critical problem in boundary value analysis is judging what is a boundary and what is not. If we could quantify how close to a boundary a given input is, we could then use a plethora of methods to find many such inputs and keep only the ones closest to a boundary. Together such a set could indicate where the boundaries of the software are or at least indicate areas closer to the actual boundaries.

In a previous work (*Feldt & Dobslaw, 2019*), we proposed the *program derivative* for quantifying boundariness. The idea is based on a generalisation of the classic definition of the derivative of a function in (mathematical) analysis. In analysis, the definition is typically expressed in terms of one point, $x$, and a delta value, $h$, which together define a second point after summation. The derivative is then the limit as the delta value approaches zero:

$$\lim_{h \to 0} \frac{f(x+h) - f(x)}{(x+h) - x} = \lim_{h \to 0} \frac{f(x+h) - f(x)}{h}$$

The derivative measures the *sensitivity to a change* in the function given a change in the input value. A large (absolute value of a) derivative indicates that the function changes a lot, even for a minimal input change. If the function $f$ here instead was the SUT, and the slight change in inputs would cross a boundary, it is reasonable that the output would also
[3]We here assume each input corresponds to a single argument of the program, but by using distance functions that can also handle multiple arguments, the formulation becomes fully general. The same holds for output $P(x)$ that may extend to any execution properties. At the same time, we limit ourselves in this study to the actual primary or secondary output (valid return value or error).

change more than if the change did not cross a boundary. We could then use this *program derivative* to screen for input pairs that are good candidates to cross the boundaries of a program.

The key to generalizing from mathematical functions to programs is to realize that programs typically have many more than one input, and their types can differ greatly from numbers. Also, there can be many outputs, and their types might vary from the types of inputs. Instead of simply using subtraction ("-") both in the numerator and denominator, we need two distance functions, one for the outputs ($d_o$) and one for the inputs ($d_i$). Also, rather than finding the closest input to calculate the derivative of a single input, for our purposes here, we only need to quantify the *boundariness* of any two individual inputs. We thus define the *program difference quotient* (PDQ) for program $P$ and inputs[3] $a$ and $b$ as *Feldt & Dobslaw (2019)*:

$$PDQ_{d_o,d_i}(a,b) = \frac{d_o(P(a),P(b))}{d_i(a,b)}, \text{ where } P(x) \text{ denotes the output of the program for input } x.$$

Since the PDQ is parameterized on the input and output distance functions, this defines not a single but a whole family of different measures. A tester can choose distance functions to capture meaningful output differences and/or inputs. In the original program derivative paper, *Feldt & Dobslaw (2019)*, we argued for compression-based distance functions as a good and general choice. However, a downside with these is that they can be relatively costly to calculate, which could be a hindrance when used in a search-based loop, potentially requiring many distance calculations. Also, compression-based distances using mainstream string compressors such as `zlib` or `bzip2` might not work well for short strings, as commonly seen in testing.

In this work, we thus use one of the least costly output distance functions one can think of: `strlendist` as the difference in length of outputs when they are printed as strings. This distance function works regardless of the type of output involved. A downside is that it is coarse-grained and will not detect smaller differences in outputs of the same length. Still, if a simple and fast distance function can suffice to detect boundaries, this can be a good baseline for further investigation. Also, our framework is flexible and can use multiple distance functions for different purposes in its different components. For example, one could use `strlendist` during search and exploration while using more fine-grained NCD-based output distance when updating the archive or during summarisation (see Fig. 1).

For the input distance function, this will typically vary depending on the SUT. We can use simple vector functions such as Euclidean distance if inputs can be represented as numbers or a vector of numbers. For more complex input types, one can use string-based distance functions like Normalised Compression Distance (NCD) (*Feldt et al., 2008*; *Feldt et al., 2016*; *Feldt & Dobslaw, 2019*) or even simpler ones like Jaccard or the related Overlap Coefficient distance (*Jaccard, 1912*).

### Example: program derivative for bytecount

We illustrate our framework with the simple `bytecount` function that is one of the most copied snippets of code on Stack Overflow but also known to contain a bug

**Table 1  Example of six boundary candidates for the bytecount SUT with their corresponding program difference quotient (PDQ) values for two different output distances.**

| Row | Input 1 | Input 2 | Output 1 | Output 2 | $d_{o1}$ (strlendist) | $d_{o2}$ (Jacc(1)) | $d_i$ | $PDQ_1$ | $PDQ_2$ |
|---|---|---|---|---|---|---|---|---|---|
| 1 | 9 | 10 | 9B | 10B | 1 | 0.75 | 1 | 1 | 0.75 |
| 2 | 999949999 | 999950000 | 999.9 MB | 1.0 GB | 2 | 0.63 | 1 | 2 | 0.63 |
| 3 | 99949 | 99950 | 99.9 kB | 100.0 kB | 1 | 0.43 | 1 | 1 | 0.43 |
| 4 | 99949 | 99951 | 99.9 kB | 100.0 kB | 1 | 0.43 | 2 | 0.50 | 0.21 |
| 5 | 99951 | 99952 | 100.0 kB | 100.0 kB | 0 | 0.0 | 1 | 0.0 | 0.0 |
| 6 | 99948 | 99949 | 99.9 kB | 99.9 kB | 0 | 0.0 | 1 | 0.0 | 0.0 |

[4]The buggy code can be found here https://programming.guide/worlds-most-copied-so-snippet.html.

(*Baltes & Diehl, 2019*; *Lundblad, 2019*).[4] bytecount is a function that translates an input number of bytes into a human-readable string with the appropriate unit, *e.g.,* "MB" for megabytes *etc*. For example, for an input of 2,099, it returns the output "2.1 kB", and for the input 9950001, it returns "10.0 MB".

Table 1 shows a set of manually selected examples of boundary candidate pairs, using a single input distance function (subtraction) but two different output distances: strlendist and Jacc(1), the Jaccard distance based on 1-grams (*Jaccard, 1912*). The Jaccard distance can approximate compression distances but is also fast to calculate and applicable for short strings. Correspondingly, in our example table, there are two different PDQ values, and we have sorted them in descending order based on the $PDQ_2$, *i.e.,* that uses the *Jacc*(1) output distance function.

Starting from the bottom of the table, on row 6, the bytecount output is the same for the input pair (99948, 99949). This leads to PDQ values of 0.0 regardless of the output distance used. The PDQ values are zero also for the example on row 5; even though the output has changed compared to row 6, they are still the same within the pair. We are thus in a different partition since the outputs differ from the ones on row 6, but we are not crossing any clear boundary, and the PDQ values are still zero.

The example in row 4 shows a potential boundary crossing. Even though the input distance is now greater, at 2, the outputs differ, so both PDQ values are non-zero. However, this example pair can be improved further by subtracting 1 from 99951 to get the pair (99949, 99950) shown in row 3. Since the denominator in the PDQ calculation is smaller, the PDQ value is higher, and we consider it a better boundary candidate. In fact, it is the input pair with the highest PDQ value of those with these two outputs; thus, it can be considered the optimal input pair to show the "99.9 kB" to "100.0 kB" boundary.

Finally, the examples on rows 2 and 1 show input pairs for which the two PDQ measures used differ in ranking. While we can agree that both of these input pairs detect boundaries, different testers might have different preferences on which one is the more preferred. Given that $PDQ_1$, using the bytecount output distance is so simple and quick to compute, we will use that in the experiments of this study. Future work can explore trade-offs in the choice of distance functions as well as how to combine them. Note that regardless of how the input pairs have been found, they can be sorted and selected using different output distance functions when presented.

## Exploration: generation of candidate pairs

While the program derivative can help evaluate whether an input pair is a good candidate to detect a boundary, there are many possible such pairs to consider. Thus, we need ways to explore the input space and propose good candidate pairs, *i.e.,* that have high program derivative values. A natural way to approach this is as a search-based software engineering problem (*Harman & Jones, 2001*; *Afzal, Torkar & Feldt, 2009*; *Feldt, 1998*), with the program derivative as the goal/fitness function and a search algorithm that tries to optimize for higher program derivative values.

However, detecting the boundaries is insufficient to find and return one candidate pair. Most software will have multiple and different boundaries in their input domain. Furthermore, boundaries are typically stretched out over (consecutive) sets of inputs. The search and exploration procedure we chose should thus output a set of input pairs that are, ideally, spread out over the input domain (to find multiple boundaries) as well as over each boundary (to help testers identify where it is).

An additional concern when using search-based approaches is the shape of the *fitness landscape*, *i.e.,* how the fitness value changes over the space being searched (*Smith, Husbands & O'Shea, 2002*; *Yu & Miller, 2006*). Many search and optimisation approaches assume, or at least benefit from, a smooth landscape, *i.e.,* small steps in the space lead to only a tiny change in the fitness value. Whether we can assume this to be the case for our problem is unclear. The program derivative might be very high right at the boundary while showing minimal variation when inside the partitions on either side of the boundary. Worst case, this could be a kind of needle-in-a-haystack fitness landscape (*Yu & Miller, 2006*) where there is little to no indication within a sub-domain to guide a search towards its edges, where the boundary is.

Given these differences compared to how search and optimisation have typically been applied in software testing, we formulate our approach in a generic sense. We can then instantiate it using different search procedures and empirically study which are more or less effective. However, given that we are searching for a pair of two inputs with a significant *derivative* in the program, we must tailor even the most basic random search baseline to this problem.

We, therefore, formulate AutoBVA detection as a generic framework with customizable components of significance. Two essential abstractions are (1) the global sampling/generation strategy to *decide starting points* for (2) the local exploration/mutation strategy to *modify those starting points in a structured way* to detect good, candidate boundary pairs. Algorithm 1 outlines this generic, 2-step automated boundary detection. Given a Software under test $SUT$, it returns a set of boundary candidate pairs ($BC$). It has three additional parameters: a way to quantify the boundariness of pairs of inputs ($Q$), a (global) sampling strategy ($SS$) to propose starting points for local exploration and a (local) boundary search ($BS$) strategy. These exploration strategies capture two different types of sampling/search procedures. The global one explores the input space of the SUT as a whole by sampling different input points (line 3). The local strategy will then search from the starting point (line 4) by manipulating it by applying *mutations*. Each of the potential new candidates (in $PBC$) is then evaluated and their boundariness is compared

to a threshold (line 5) and added to the final set ($BC$) returned. The `bytecount` function simply captures the fact that we might not use a fixed threshold but rather can allow more complex updating schemes where the threshold is based on the candidates that have already been found. For example, the threshold could be taken as some percentile (say, 90%) of the boundariness values of the candidate set saved so far. Alternatively, even more, elaborate boundariness testing and candidate set update procedures can be used, such as rather than simply adding to the current set whenever a sufficiently good boundary value is found, we could save a top list of the highest boundariness values found so far. The update would then be generalized so that it can also delete previously added candidate pairs that are no longer promising.

---

**Algorithm 1** Automated Boundary Detection - AutoBD-2step

---

**Input:** Software under test $SUT$, Boundariness quantifier $Q$, Sampling strategy $SS$,
   Boundary search $BS$
**Output:** Boundary candidates $BC$

1:  $BC = \emptyset$
2:  **while** stop criterion not reached **do**
3:     $input = SS.sample(SUT, BC)$ # globally sample a starting point
4:     $PBC = BS.search(SUT, Q, input)$ # locally explore and detect potential candidate(s)
5:     $BC = BC \cup \{c | c \in PBC \wedge Q.evaluate(c) > threshold(BC)\}$
6:  **end while**
7:  **return** $BC$

---

The sampling strategy defines *where* we start looking, *i.e.,* what input we start our search from. The local search strategy defines *how* we manipulate that input to identify boundary candidates. For complex data types, such as XML trees or images, more sophisticated generators are required. For numbers, the shelf sampling from basic distributions, such as uniform at random, suffices. We use uniform at-random sampling as a sampling baseline.

To illustrate, consider a $SUT$ that takes a single integer number as input. The sampler returns a random number, say 10. Let us assume that the local search has access to increment and decrement mutations. In the most naive form, the local search simply applies each mutation operator once per argument to obtain a neighbourhood of distance one. The candidate pairs $(9, 10)$ and $(10, 11)$ would be probed in the local search. If the $SUT$ in question were actually the above-mentioned `bytecount` PBC, it would then return all candidates above a threshold - here, the candidate pair $(9, 10)$ since it has the highest PDQ value. Since this is among the simplest mechanisms for deriving boundary candidates, it informed our formulation of the Local Neighbourhood Search below, which we use as a baseline.

The benefit of creating candidate input pairs in two steps may become particularly obvious when the inputs are of a complex and structured datatype, such as an XML tree. If search was done with a single, global exploration step, it would be relatively unlikely that the two inputs of a candidate pair would have a small distance or that there even is a useful way of generating inputs that lead from one point to the other. By doing this in two

steps, we could first sample XML tree instances and then explore their neighbourhoods in the input space by small mutations. This mechanism, however, is not limited to exploring only the very near neighbourhood of a point but may be used to explore the space by a repeated application of mutation operators. This idea informed our second local search strategy. Future work may still look into using global sampling only. Still, the combination of arbitrary inputs to detect boundaries seems non-trivial and might require tailored crossover mechanisms for input types.

For this study, we first benchmark the impact of the global sampling (uniform, bituniform) in a screening study to understand its effect on the results. The study is presented in Appendix A. We found that uniform sampling (often referred to as random sampling) performed substantially worse-in fact, it did not find any boundary candidates in any attempt when another feature was deactivated. Consequently, we simplified the experimental design of the main study to bituniform sampling. Next, we explain the two boundary search algorithms mentioned above: Local Neighbour Sampling and Boundary Crossing Search.

### Global sampling

As mentioned above, the initial implementation of AutoBVA for validation of overall applicability is limited to numbers. By our findings on poor performance in the screening (see Appendix A) for uniform (random) sampling but promising performance for bituniform sampling, we deem uniform sampling insufficient as a sampling baseline (Algorithm 1, line 3). We hypothesise that this is because it favours larger numbers not covering the entire input spectrum-and arguably less of the often more interesting regions, including the switch between positive and negative numbers around zero. Bituniform sampling selects random numbers with a bit-shift that inserts leading zeros. The number of leading zeros is decided uniformly at random in the range of 0 (no change to the orignal number) and the bit-size of the datatype (*e.g.*, 64 for 64 bit integers).

For a broad exploratory sampling, we introduce a complementing technique that we call compatible type sampling (CTS), *i.e.,* argument-wise sampling based on compatible types per argument. An example of a compatible type is any integer type with a specific bit size. For instance, in the Julia programming language, we use in our experiments, booleans (Bool), 8-bit (Int8), and 32-bit integers (Int32) types are compatible because they all are sub-types of integer.

More details and justification for the global sampling strategy we use here, bituniform sampling combined with CTS, can be found in Appendix A. We do note that in general, more advanced test input generation strategies (*Feldt & Poulding, 2013*) can be used, and adapting them to the SUT and its arguments will likely be important when further generalizing our framework. We revisit the topic of handling more complex software in 'Discussion'.

### Local neighbour sampling

Because we are searching for pairs, among the simplest imaginable strategies for local search in Algorithm 1 (line 4), local neighbourhood sampling (LNS), is presented as Algorithm 2. The basic idea with LNS is to structurally sample neighbouring inputs, *i.e.,*

inputs close to a given starting point $i$, to form potential candidate pairs, including $i$. The algorithm processes mutations over all individual arguments (line 3), considering all provided mutation operators $mos$ (line 4). For integer inputs, the mutation operators are basic subtraction and addition (of 1). Outputs are produced for the starting point (line 2) and each neighbour (line 6) to form the candidate pairs (line 7). Without filtering by, *e.g.*, a program derivative, they are all *blindly* added to the set of potential boundary candidates (line 8), returned by the algorithm (line 11). LNS is a trivial baseline implementation to understand better what is possible using AutoBD without a sophisticated search. LNS will invariably return pairs that contain starting point $i$ as one side of the boundary candidate. We include this method as a baseline to understand whether a more sophisticated and time-consuming method is justified.

---

**Algorithm 2** *search* – Local Neighbour Sampling (LNS)

---

**Input:** Software under test $SUT$, mutation operators $mos$, Starting Point $i$

**Output:** potential boundary candidates $PBC$

---

1:  $PBC = \emptyset$
2:  $o = SUT.execute(i)$
3:  **for** $a \in arguments(SUT)$ **do**
4:    **for** $mo \in mos[a]$ **do**
5:      $n = mo.apply(i, a)$
6:      $o_n = SUT.execute(n)$
7:      $c = \langle i, o, n, o_n \rangle$
8:      $PBC = PBC \cup \{c\}$
9:    **end for**
10:  **end for**
11: **return** $PBC$

---

[5] A stop criterion returns the original point in case no difference could be picked up.

### Boundary crossing search

Boundary crossing search (BCS) is a more sophisticated heuristic local search strategy (see Algorithm 3) that uses a boundariness quantifier $Q$, in our experiments the program derivative. BCS seeks a locally derived potential boundary candidate pair that stands out compared to starting point $i$. For a random direction (argument $a$ in line 1 and mutation operator $mo$ in line 2) a neighbouring input $i_{next}$ gets mutated (line 3) and outputs for both inputs are produced (line 4) to define the initial candidate (line 5) for which the boundariness gets calculated (line 6).

Lines 7–12 describe the constraints and conditions for a resulting boundary candidate $c$ to stand out locally. This search can be implemented in a variety of ways. We implemented a binary search that first identifies the *existence* of a boundary crossing by taking ever greater steps and calculating the difference $\Delta_c$ to find an input for the state in which the boundariness is greater than $\Delta_{init}$, thereby guaranteeing the necessary condition in line 12.[5] Once that is achieved, the algorithm *squeezes* that boundary to obtain $c$, which is the
[6]In practice for numbers, this is implemented by subtraction of a larger constant 2, 4, *etc.* but the framework abstracts this as a concatenation of mutation operators and is therefore extendable to any types of data types and mutations.

[7]It can be noted that because of the binary nature of the search, the distance to the original point 13 is first one (12), then two (10), then four (6), and so forth.

---

**Algorithm 3** *search* – Boundary Crossing Search (BCS)

**Input:** Software under test $SUT$, Boundariness quantifier $Q$, mutation operators *mos*, Starting Point $i$

**Output:** potential boundary candidates $PBC$

1:   $a = rand(arguments(SUT))$ # select random argument

2:   $mo = rand(mos[a])$ # select random mutation operator for argument

3:   $i_{next} = mo.apply(i, a)$ # mutate input a first time in single dimension

4:   $o = SUT.execute(i), o_{next} = SUT.execute(i_{next})$ # produce outputs

5:   $c_{init} = \langle i, o, i_{next}, o_{next}\rangle$ # instantiate initial candidate

6:   $\Delta_{init} = Q.evaluate(c_{init})$ # calculate candidate distance

7:   $c = \langle i_1, o_1, i_2, o_2 \rangle$, with $i_1$ obtained by a finite number of chained mutations *mo* of *a* over *i*, and

8:                      $i_2 = mo.apply(i_1, a)$, and

9:                      $o_1 = SUT.execute(i_1)$, and

10:                    $o_2 = SUT.execute(i_2)$, and

11:                    $\Delta_c = Q.evaluate(c)$, and

12:                    $\Delta_c > \Delta_{init}$ or $c_{init} == c$

13: **return** $\{c\}$

---

nearest point to $c_{init}$ that ensures a greater local difference in neighbouring inputs, and by that guarantees the neighbouring constraint in line 8.

We illustrate the search by finding a date (three integer inputs) boundary candidate with starting point *thirteenth of February 2022*, or $(13, 2, 2022)$, and using the program derivative with bytecount as output metric. Let us further assume that argument 1 or *day* in combination with mutation operator subtraction are randomly selected. We probe and get $Date(i) =$ "2022-13-02". BCS will probe the initial neighbour $mo(i) = i_{next} = (12, 2, 2022)$, which results in "2022-12-02" and no conceivable difference, with $\Delta_{init} = 0$. The algorithm will apply the mutation operator in a binary fashion, meaning first twice $mo(mo(i_{next}))$, then four times, and so on until it perceives a larger difference.[6] BCS thus first tries days 10 and then 6 with no perceivable difference still[7] Next, at -2 it perceives a difference due to an error message that signals the date out of bounds. From here on, it backtracks to *squeeze* the boundary to identify the two neighbouring points for which the transition happens, as guided by the program derivative. There are multiple strategies to do this, and a simple one is to apply binary cuts between the two points to eventually find the candidate pair $\langle(0, 2, 2022), (1, 2, 2022)\rangle$ for which the input distance is atomic, and the program derivative locally maximal (in the starting point, argument, and mutation operator). In the general case, these detected potential boundaries could both separate two valid outputs, as well as a combination of valid and error outputs (such as in our case), or even two error cases. As explained in the following section, we use this grouping information in the boundary candidate summarisation.

## Summarisation: validity-value similarity clustering

The boundary candidate set resulting from AutoBD-2step (Algorithm 1) can be extensive. However, human information processing is limited, and more fundamentally many of the boundary pairs found can be similar to each other and represent the same or a very similar boundary. Take, for instance, the Date example with a great number of different boundary pairs crossing the same behavioural boundary between valid dates and the invalid 0 for the day - $\langle (0, m, y), (1, m, y) \rangle$. The summary method(s) we choose should thus not only limit the number of candidates presented, but those candidates also need to be different and represent different groups of behaviours and boundaries over the input space.

Furthermore, the goals for the summary step will likely differ depending on what the boundary candidates are to be used for; comparing a specification to actual boundaries in an implementation is a different use case than adding boundary test cases to increase coverage. Thus we cannot provide one general solution for summarisation, and future work will have to inspect different methods to cluster, select, and prioritize but also visualize the boundary candidates.

In the following, we propose one particular but very general summarisation method. We hope it can show several building blocks needed when creating summarisation methods for specific use cases and act as a general fallback method that can provide value regardless of the use case. This is also the method we use in the experimental evaluation of the article. We consider a general, black-box situation with no specification available. Thus, we only want to use information about the boundary candidates themselves. The general idea is to identify meaningful clusters of similar candidates, then sample only one or a few representative candidates per cluster and present them to the tester.

For instance, Table 2 contains a subset of (nine) candidate pairs found by our method for the `bytecount` example introduced above. Different features can be considered when grouping and prioritizing. We see that candidates differ at least in the output type, *i.e.*, whether the outputs are valid return values or exceptions, as well as in the actual values of inputs and/or outputs themselves. For example, both outputs for the candidate on row 2 are strings and are considered normal, valid return values. On the other hand, for row 9, both outputs are (Julia) exceptions indicating that a string of length 6 ("*kMGTPE*") has been accessed at (illegal) positions 9 and 10, respectively.

We argue that the highest level boundary in boundary value analysis is between valid and invalid output values. Any time there is an exception thrown when the *SUT* is executed, we consider the output to be invalid; if not, the output is valid.[8] Since we consider pairs of inputs, two outputs per candidate can be grouped into three, what we call, *validity groups*:

- **VV:** two valid outputs in the boundary pair.
- **VE:** one valid output and one invalid (error) output.
- **EE:** two invalid (error) outputs.

We use validity as the top-level variation dimension and produce individual summaries for these three groups. Table 2 indicates the validity group in the rightmost column (VV for lines 1–7, VE in line 8, and EE in line 9). We may consider other characteristics within each validity group to categorize candidates further. For example, we could use the type of

[8]Note that in the implementation, we have to clearly distinguish between the situation when an exception was thrown during execution from the one where the function itself returns a value that is an exception. Otherwise, our framework could not be used for functions that manipulate exceptions (without raising any exceptions).

**Table 2  A summary of boundary candidates found for bytecount by our method.** Rows 1–7 are for the valid-valid (VV) validity group of type String-String, row 8 for the valid-error (VE) group of type String-BoundsErrror, and row 9 for error-error (EE) group of type BoundsError-BoundsError. The candidates of rows 1–7 are the shortest candidate pairs of each identified cluster.

| Row | Input 1 | Input 2 | Out 1 | Out 2 | Validity |
|---|---|---|---|---|---|
| 1 | false | true | falseB | trueB | VV |
| 2 | 9 | 9B | 10 | 10B | VV |
| 3 | −10 | −9 | −10B | −9B | VV |
| 4 | 999949 | 999950 | 999.9 kB | 1.0 MB | VV |
| 5 | 99949999999999999 | 99950000000000000 | 99.9 PB | 100.0 PB | VV |
| 6 | 9950000000000001999 | 9950000000000002000 | 9.9 EB | 10.0 EB | VV |
| 7 | −1000000000000000000000000000000 | −999999999999999999999999999999 | −1000000000000000000000000000000B | −999999999999999999999999999999B | VV |
| 8 | 999999999999994822656 | 999999999999994822657 | 1000.0 EB | BoundsError("kMGTPE", 7) | VE |
| 9 | 9999999999999990520104160854016 | 9999999999999990520104160854017 | BoundsError("kMGTPE", 9) | BoundsError("kMGTPE", 10) | EE |

the two outputs to create additional (sub-)groups. This is logical since it is not clear that comparing the similarity of values of different types is always meaningful. However, for many SUTs and their validity groups, the types might not provide further differentiation as they often are the same.

In the final step, we propose to cluster based on the similarity of features within the identified sub-groups. A type-specific similarity function can be used, or a general, string-based distance function can be used after converting the values to strings. We can create a distance matrix per sub-group after calculating the pair-wise similarities (distances). This can then be used with any clustering method while in the experiments in this article, we used k-means clustering (*Likas, Vlassis & Verbeek, 2003*). We select one representative (or short) boundary pair from each cluster to present to the tester. The distance matrix can also be used with dimensionality reduction methods to visualize the validity-type sub-group. We thus call our summarisation method *validity-value similarity clustering*.

The duality of having two sites of a boundary exposes more structure than thinking in test cases as single points with the expected outcomes only. Therefore, the validity groups are not meant to be a sufficient differentiator but a tool to exploit general structure for better separation for most, if not all SUTs.

## EMPIRICAL EVALUATION

This section starts by explaining the overall aim of the experimental study-broken up into research questions RQ1–RQ3-with its two investigations. Details about the experimental design and setup can be found in 'Selection: Program Derivative', with subsections for each investigation. 'Setup of summarisation step' then describes the applied clustering approach with the utilized metrics that define the feature space for the summarisation in both investigations.

We run our framework using the two local search strategies, LNS and BCS, introduced above (independent variable) in two investigations and study the sets of boundary candidates returned in detail. Investigation 1 offers a deep analysis of four curated SUTs of different types, whereas Investigation 2 tests the applicability of all compatible functions in Julia's core library called Base (control variables). All SUTs are program functions.

We evaluate the degree to which the framework can find many diverse and high-quality candidate boundary pairs. Specifically, we address the following research questions:

- RQ1: Can AutoBVA identify large quantities of boundary candidates?
- RQ2: Can AutoBVA robustly identify boundary candidates that cover a diverse range of behaviours?
- RQ3: To what extent can AutoBVA reveal relevant behaviour or potential faults?

Through RQ1, we try to understand to what extent AutoBVA can pick up *potential* boundary candidates (PBC) by comparing two local search strategies. We analyse the (1) overall quantities and (2) quantities of *uniquely* identified PBCs using a basic boundary quantifier. The uniqueness here is measured in relation to the set of all PBCs for a SUT over all repetitions of our experiments, irrespective of the local search applied.

Through RQ2, we try to understand how well AutoBVA covers the space of possible boundaries between equivalence partitions in the input space of *varying* behaviour. For a given arbitrary SUT, we argue that there is no one-size-fits-all approach to extracting/generating "correct" equivalence partitions; many partitions and, thus, boundaries exist and which ones a tester considers might depend on the particular specification, the details with which it has been written, the interests and experience of the tester, *etc.* Therefore, we use our summarisation method (validity-value similarity clustering as described in 'Summarisation: validity-value similarity clustering') to group similar PBCs within each validity group (VV, VE, and EE) and apply clustering per each group. We answer RQ2 by analysing how the PBCs found by each exploration method cover those different clusters. Comparing the coverage of these clusters allows us to interpret the behaviour triggered by the set of PBCs. For instance, the two boundary candidates identified for the Date constructor SUT, bc1 = (**28**-02-2021, **29**-02-2021) and bc2 = (**28**-02-2022, **29**-02-2022) are different PBCs, but they do cover the same valid-error boundary.[9] It is, therefore, not clear that finding those two specific PBCs, or many similar boundary candidates showing a similar boundary, helps identify diverse boundary behaviour.

The quantitative approaches used for RQ1 and RQ2 cannot probe whether the candidates found are high quality, *i.e.,* if the boundaries they indicate are unexpected or essential to test. For RQ3, we thus perform a qualitative analysis of the identified PBCs by manually investigating each cluster, systematically sampling candidates with varied output lengths, and analysing them. Consequently, we examine whether and how often AutoBVA can identify relevant, rare, and/or fault-revealing SUT behaviour.

A sub-question for all three RQs is how the different local search/exploration strategies compare against one another in detecting unique boundary candidates. In short, the dependent variables in our experiment are the number of (unique) candidates found (RQ1), the number of (unique) clusters covered (RQ2), and the characteristics of interesting candidates (RQ3) found by each exploration approach. Next, we detail how we set up each stage of AutoBVA in our experiments.

[9]Note that both pairs belong to the valid-error (VE) validity group as both 2021 and 2022 are **not** leap years, and thus February 29 leads to an ArgumentError exception being thrown.

**Table 3  The different configurations used to run each of the four SUTs.**

| Parameter | Investigation 1 | Investigation 2 |
|---|---|---|
| Sampling method (SS) | bituniform + CTS activated | same as Investigation 1 |
| Exploration strategy (ES) | LNS, BCS | same as Investigation 1 |
| Boundariness quantifier (PD) | strlendist | same as Investigation 1 |
| Threshold | 0 | same as Investigation 1 |
| Mutation operators (m) | increment/decrement (++/–) | same as Investigation 1 |
| SUTs | bytecount, BMI-Value, BMI-Class, Date | 580 functions (Julia Base) |
| Stop criterion | {30,600} seconds | 30 seconds |

## Setup of selection and exploration step

Before the main experiment, we performed two screening studies to configure (1) the (global) sampling strategy and (2) the clustering of boundary candidates (see Appendices A and B). Table 3 summarises the setup of both investigations. The sampling method is fixed as the best-performing configuration from the screening study and uses both bituniform sampling and CTS for all experiments. The boundariness quantifier is based on the program derivative with bytecount for the outputs, *i.e.,* the length difference between the stringified versions of outputs. Since all input parameters of the SUTs in this study are integer types, numerical distance is used (implicitly) as input distance. In the search, we use an increment (add to integer) and a decrement (subtract from integer) as mutation operators. We repeat each experiment execution 20 times to account for variations caused by the pseudorandom number generator used during the search. A constant and permissive threshold of 0 for adding boundary candidates to BC is applied in this study, *i.e.,* all pairs with any difference in output length are added. Setup specific for each investigation is taken up below.

### *Investigation 1*

For each SUT, we conducted two series of runs, one short for 30 s each and one longer for 600 s (10 min), to understand the convergence properties of AutoBVA. We selected those two time-limits to loosely assess a tester's more direct (30 s) or offline (10 min) usage.

We investigate four SUTs: a function to print byte counts, Body Mass Index (BMI) calculation as value and category, and the constructor for the type Date. The SUTs have similarities (*i.e.,* unit-level that have integers as input), but each has peculiar properties and different sizes of input vectors. For instance, when creating dates, the choice of months affects the day's range validity (and vice-versa), whereas the result of a BMI classification depends on the combination of both input values (height and weight). Below, we explain the input, output, and reasoning for choosing each SUT. The code for each SUT is available in our reproduction package (https://doi.org/10.5281/zenodo.7677012).

bytecount (i1: Int): Receives an integer representing a byte value (i1). The function returns a human-readable string (valid) or an exception, signalling whether the input is out of bound (invalid). The largest valid inputs are those represented as Eta-bytes. We chose this SUT because it is the most copied code in StackOverflow. Moreover, the code faults the boundary values when switching between scales of bytes (*e.g.,* from 1,000 kB to 1.0 MB).

`bmi_value (h: Int, w: Int)`: Receives integer values for a person's height ($h$, in cm) and weight ($w$, in kg). The function returns a floating point value resulting from $w/(h/100)^2$ (valid) or an exception message when specifying negative height or weight (invalid). The SUT was chosen because the output is a ratio between both input values, *i.e.,* different height and weight combinations yield the same BMI values.

`bmi_classification (h: Int, w: Int)`:: Receives integer values for a person's height ($h$, in cm) and weight ($w$, in kg). Based on the result of `bytecount`, the outcome is a category string serving as a health indicator with six categories, spanning from underweight all the way to severely obese in the valid range, and causing exceptions if called with negative height or weight values. This design was chosen because the boundaries between classes depend on the combination of the input values, leading to various valid and invalid combinations.

`date (year: Int, month: Int, day: Int)`: Receives an integer value representing a year, month, and day. The function returns the string for the specified date in the proleptic Gregorian calendar (valid).[10] Otherwise, it returns specific exception messages for incorrect combinations of year, month, and day values (invalid). The Date SUT was chosen because it has many boundary cases conditional to the combination of outputs (*e.g.*, the maximum valid day value varies depending on the month or the year during a leap year).

Our choice of SUTs offers a gradual increase in the input complexity, where the tester needs to understand (1) individual input values, (2) how they will be combined according to the specification, and (3) how changing them impacts the behaviour of the SUT. For instance, when choosing the input for the year in the date constructor, a tester can choose arbitrary integer values (case '1') or think of year values that interact with February 29 to check leap year dates (case '2'). Another example would be choosing a test input for BMI, in which the tester needs to manually calculate specific height and weight combinations to verify all possible BMI classifications (case '3'). A tester must check the boundaries for the types used (*e.g.*, maximum or minimum integer values) in parallel to all those cases. Note that systematically thinking of inputs to reveal boundaries is a multi-faceted problem that depends on the SUT specification (*e.g.*, what behaviours *should* be triggered), the values acceptable for the input type and the output created independently of being a valid outcome or an exception.

### Investigation 2

To investigate AutoBVA's generalizability and mitigate selection bias in a realistic scenario, we test it on a non-curated set of actual SUTs (the base module in Julia). Unlike Investigation 1, we limit the execution time per SUT to 30 s as we mostly want to understand whether interesting candidates can be gathered.

The base module in Julia 1.8.0 contains a wide range of basic services that all Julia programs can access by default. We run AutoBVA in 580 functions out of 2,375 (24.4%). In Julia, each function (name) yields several methods that, in turn, have unique signatures (name + input parameters). For instance, the `bytecount` function comes with four methods, of which each is considered a unique SUT in our study to obtain a more

[10] We use the constructor from the Julia language as in https://docs.julialang.org/en/v1/stdlib/Dates/.

fine-grained detection assessment. Ultimately, the 580 functions resulted in 613 SUTs investigated in our experiment.

The resulting sample of 613 methods is divided between 259 explicitly exported methods and 358 non-exported ones. Non-exported methods can be used in Julia with the globally unique full namespace qualifier. We could not run AutoBVA on all functions because (1) our current framework implementation only supports functions with inputs of type integer, and (2) we sample only functions with at most three input parameters. To support future research studies aiming to also automatically sample from Julia Base functions, we share the following obstacles that we faced:

- Circa 20 exported functions were pre-filtered as they were deemed too low-level (operators such as and, or, and multiplier that might have direct representations in the command set of the CPU).
- A set of 36 functions crashed the AutoBVA due to memory allocation issues for large integer inputs. Another special case that crashed AutoBVA was when it used, as a SUT, the `exit` function that exits Julia.

Numbers for LNS and BCS are reported separately to highlight their respective impact for very short runtimes, which may benefit the more straightforward strategy. RQ1 is addressed by reporting the proportion of functions for which boundary candidates could be detected. Average and standard deviation measures are offered for the quantitatively successful ones, *i.e.,* those for which candidates could be detected. RQ2 is firstly addressed quantitatively by the number of successfully summarised SUTs. A SUT is successfully summarised if a mean Silhouette score of above 0.6 is reported for all its clusterings in the validity groups-a value that is commonly recognized as an acceptable/good clustering. Secondly, (unique) cluster coverage is reported in an aggregate per exploration strategy. RQ3 is addressed by observations about success factors in successful SUTs and potential reasons for issues that did not lead to successful detection. This will be exemplified by several non-successful ones and their commonalities and a selection of the two successful ones with the best Silhouette scores that are analyzed similarly to the four SUTs in Investigation 1 through cluster representative tables.

### Setup of summarisation step

To summarise a large set of boundary candidates, we have to decide and extract a set of complementing features able to group similar candidates and set apart those that are dissimilar. Ideally, we want a generic procedure which can give good results for many different types of SUTs. We want to select features so boundary pairs with similar outputs are grouped. The focus of this study has been on detection, while the following reasoning regarding simplicity and feature selection guided the applied summarisation.

We implement the AutoBVA summarisation by validity-value similarity clustering using k-means clustering *Hartigan & Wong (1979)*. We choose k-means clustering because it is one of the simplest, well-studied, and understood clustering algorithms and widely available *Xu & Tian (2015)*. Clustering was done per validity group to avoid mixing pairs that have very different output types, namely: VV (String, String), VE (String, Error), and EE (Error,

Error). We span a feature space over the boundary candidates to capture a diverse range of properties and allow for a diversified clustering from which to sample representatives per cluster. We extract features from the output differences between boundary candidates since input distances within the boundary candidates are already factored into the selection of the candidates. Moreover, outputs can easily be compared in their "stringified" version using a generic information theoretical measure $Q$, typically a string distance function.

Our goal is that the features that span the space shall be generic and capture different aspects of the boundary candidates. We, therefore, introduce two feature types: (1) WD captures the *differences within* a boundary candidate as the distance between the first and the second output; (2) U is a two-attribute feature that captures the *uniqueness* of a candidate based on the distance between the first ($U_1$) and second ($U_2$) output to the corresponding outputs of all other candidates in the set.[11] Considering $Q$ as the distance measure chosen for the outputs, we define U and WD for a boundary candidate $j \in BC, j = (out_{j1}, out_{j2})$ as:

$$WD_j = Q(j_1, j_2)$$

[11] All distance values from $Q$ are normalised between zero and one to keep all features and corresponding attributes on the same scale.

[12] For occasional comparisons with single character strings, this defaults back to Overlap of length one as it otherwise leads to divisions by zero.

$$U_j = (U_{j1}, U_{j2}); \text{ where } U_{jk} = \sum_{j' \in BC} Q(j_k, j'_k), \text{ k} = 1, 2$$

To understand which combination of distance measures ($Q$) yields better clustering of boundary candidates, we conducted a screening study using three different string distances to measure the distance between the outputs, namely, `strlendist`, Levensthein (`Lev`) and `Overlap` Coefficient of length two.[12] These common metrics cover different aspects of string (dis)similarity, each with its own trade-off. For instance, `strlendist` is efficient but ignores the characters of both strings, whereas `Overlap` compares combinations of characters but disregards specific sequences in which those characters show up (*e.g.*, missing complete names or identifiers); lastly, `Lev` is the least efficient but more sensitive to differences between the strings. Nonetheless, all three measures only consider lexicographic similarity and are not sensitive to semantics, such as synonyms or antonyms.

For simplicity, the screening study was done only on the `bytecount` SUT. Details of the screening study and examples of features extracted from boundary candidates are presented in Appendix B. Our screening study reveals that `strlendist`(WD) and `Overlap`(WD, U) is the best combination of features and distance measures that yield clusters of good fit with high discriminatory power. Choosing those types of models yields more clusters that can be differentiated with high accuracy, hence allowing for a more consistent comparison of cluster coverage between exploration strategies. Moreover, clearer clusters are also useful in practice, allowing AutoBVA to suggest testers with more diverse individual boundary candidates.

Formally, we create a feature Matrix $M$ over boundary candidates of a SUT with each row $i$ representing each attribute from the features over each boundary candidate $j$, as defined below.

$$M = \begin{bmatrix} WD \\ U \end{bmatrix},$$

For this experiment, each $M$ has four rows, one per attribute in the chosen features: `strlendist(WD)`, `Overlap(WD)`, `Overlap`($U_1$) and `Overlap`($U_2$). The number of columns ($j$) varies depending on the number of candidates found per exploration approach and SUT. Since k-means clustering is a heuristic algorithm, we run the clustering 100 times on each SUTs feature matrix $M$ to retain the model of clustering of best fit according to the Silhouette score. To evaluate the coverage of the boundary candidates (RQ2), we choose the clustering discriminating best, *i.e.,* the one resulting in the most clusters based on the top five percentile Silhouette scores.

We improve overall clustering quality by selecting only a diverse subset of boundary candidates for the clustering. For that, we create an initial diversity matrix as of above using 1,000 randomly selected candidates.[13] We then substitute the least diverse 100 candidates (based on the sum of all normalized diversity readings) with 100 of the remaining candidates. Until no more candidates exist, we repeat this step to receive M for clustering. In the second step, we assign all candidates not part of M to the cluster with the closest cluster centre. A positive side-effect of this procedure is that it is much more memory efficient (limiting matrix sizes roughly to 1,000 candidates × 4 features).

[13]For SUTs with fewer than that, we skip this procedure entirely.

# RESULTS AND ANALYSIS

We here present the results and analyze them in correspondence to RQ1–RQ3 ('RQ1-Boundary candidate quantities, RQ2-Robust coverage of diverse behaviours, RQ3-Identifying relevant boundaries'). The answers combine Investigations 1 and 2, and each section concludes with a list of key findings. RQ3, with its focus on relevant behaviour and potential faults, demands a thorough analysis, which is why it is subdivided into a subsection per SUT for Investigation 1 ('Bytecount, BMI classification'), a subsection for Investigation 2 on Julia Base ('RQ3 and Investigation 2'), and an RQ3 summary subsection (Summary for RQ3).

## RQ1-Boundary candidate quantities

Table 4 summarises the number of common and unique boundary candidates found by the two search strategies, LNS and BCS of Investigation 1. For each SUT and search strategy, it shows results for the 30-second and the 600-second runs individually. For each time control, the mean and standard deviation, over the 20 repetitions, of the number of potential boundary candidates as well as the number of unique candidates, is listed. For example, we can see that BCS in a 600-second run for the `Date` SUT finds, on average, 897.4 +/- 82.6 PBCs out of a total of 45456 unique ones (found over all runs and search strategies). And overall, 7,276 of the total 45,456 PBCs were uniquely only found by BCS, *i.e.,* in none of the LNS runs, any of these 7,276 were found.

We see that overall, AutoBVA produces a large number of boundary candidates with either exploration strategy. Except for `bytecount`, there is also a large increase in the number of candidates found as the execution time increases. While the number of candidates found per second does taper off also for `BMI-class`, `BMI` and `Julia Date` (from 166, 552, and 81 per second for the short runs to 117, 380, and 76 for the long runs, respectively), longer execution clearly has the potential to identify more candidates. Since

**Table 4  Descriptive statistics (μ ± σ) over the potential boundary candidates (PBC) found by both BCS and LNS.**  Total refers to the size of the union set of candidates found during the 20 executions of each strategy.

| SUT | Strategy | 30 seconds | | | 600 seconds | | |
|---|---|---|---|---|---|---|---|
| | | Total | # PBC found | # Unique | Total | # PBC found | # Unique |
| bytecount | LNS | 57 | $10.05 \pm 0.8$ | 0 | 59 | $12.8 \pm 0.8$ | 0 |
| | BCS | 57 | $56.7 \pm 0.5$ | 44 | | $57.85 \pm 0.4$ | 43 |
| BMI class | LNS | 25,207 | $1,747.85 \pm 86.6$ | 23,238 | 358,956 | $24,421.7 \pm 354.9$ | 332,120 |
| | BCS | 25,207 | $149.85 \pm 13.3$ | 1,276 | | $1,944.8 \pm 74.5$ | 18,358 |
| BMI | LNS | 90,027 | $6,147.45 \pm 95.1$ | 86,157 | 1,280,955 | $87,319.0 \pm 1747.8$ | 1,226,314 |
| | BCS | 90,027 | $272.7 \pm 15.8$ | 2,481 | | $3873.05 \pm 124.4$ | 36,186 |
| Julia Date | LNS | 2,444 | $246.1 \pm 13.7$ | 2,232 | 45,456 | $4,351.6 \pm 86.3$ | 37,216 |
| | BCS | 2,444 | $21.6 \pm 5.9$ | 191 | | $897.4 \pm 82.6$ | 7,276 |

**Table 5  A summary of Investigation 2 for boundary detection shows the number of SUTs for which boundaries were successfully detected and how many per run (μ±σ) were found, divided by exploration strategy.**

| Exported | Strategy | # SUTs | # SUTs success (%) | # PBCs per run (30 s.) |
|---|---|---|---|---|
| yes | LNS | 259 | 197 (76%) | $1,042 \pm 76$ |
| | BCS | | 198 (76%) | $554 \pm 40$ |
| no | LNS | 354 | 243 (69%) | $578 \pm 67$ |
| | BCS | | 247 (70%) | $391 \pm 37$ |

the 20 times longer execution time for bytecount only finds one additional candidate (58 total *versus* 57 for 30 s), it might be useful to terminate search and exploration when the rate of new candidates found goes below some threshold.

For the bytecount SUT, only BCS finds a unique set of candidates, meaning that BCS also identified all boundaries identified by LNS. This means that only 14 (58–44) of all candidates found were found by LNS, even after 20 runs of 600 s, and BCS also found all of those. For the other SUTs, LNS clearly finds more candidates and more unique candidates, between 5 and 15 times more, depending on the SUT.

The boundary detection statistics for Investigation 2 are summarised in Table 5. Boundary candidates could be identified with both exploration strategies in roughly 76% of the exported SUTs and 70% of the non-exported SUTs. Even though BCS could cover more SUTs in numbers, the difference is not statistically significant. LNS detects significantly more candidates.

Thus, overall, LNS produces a higher quantity of boundaries. This is expected since it is a random search strategy with minimal local exploration. The effect can likely be explained by two reasons that may also interplay. First, the low algorithmic overhead of the LNS search method enables it to make more calls to the underlying SUT, given a fixed time budget. Second, a proportion of the BCS searches can fail and return no boundary candidates since the input landscape does not regularly lead to changes in output partition through single-dimensional mutations, *i.e.,* to one input. However, the quantity

**Table 6  Summary of the resulting clusters per validity groups: VV, VE, and EE.**

| | # of Clusters | | | # of Pairs | | | Silhouette Score | | |
|---|---|---|---|---|---|---|---|---|---|
| SUT | VV | VE | EE | VV | VE | EE | VV | VE | EE |
| bytecount | 6 | 1 | 1 | 57 | 1 | 1 | 0.95 | 1.0 | 1.0 |
| BMI | 3 | 3 | – | 579,030 | 759,380 | 0 | 0.82 | 0.0 | – |
| BMI-class | 13 | 2 | – | 7,288 | 367,125 | 0 | 1.0 | 1.0 | – |
| Julia Date | 7 | 2 | 3 | 368 | 77,207 | 253,875 | 0.73 | 0.89 | 0.89 |

of boundary candidates might not directly translate to finding more diverse and "better" boundary candidates; next, we thus consider RQs 2 and 3 addressing this issue.

> **Box 1.** *Key findings (RQ1)*:
>
> Both exploration strategies can identify large quantities of boundaries. Overall, Local Neighbour Search (LNS) finds more and more unique candidates than Boundary Crossing Search (BCS) for the more complex SUTs with multiple input arguments. In contrast, BCS finds larger numbers and more unique candidates for the one-input SUT. AutoBVA could successfully detect boundaries in over 70% of the investigated SUTs.

## RQ2-Robust coverage of diverse behaviours

Using our validity-value similarity clustering summarisation method, we obtained between six and 15 clusters in Investigation 1 across the different SUTs and validity groups (VV, VE and EE) with high clustering quality scores (see Table 6). We see that most clusters were differentiated in the VV group. No EE candidates were identified for BMI and BMI-class because of the lack of detected boundary candidates. We also find that, except for the clustering for BMI in group VE, all attempted clusterings had a high discriminate score.

We summarise the coverage of clusters per SUT, exploration strategy, and execution time in Table 7. For example, we can see that for the Julia Date SUT, after we merged all candidates found by any of the methods in any of the runs and clustered them, we found 11 clusters. In a 30-second run, LNS covered 4.9 +/- 0.3 of them and covered one cluster that was not covered by BCS (in a 30-second run), while in a 600-second run, BCS covered 7.5 +/- 0.8 clusters and covered six that were not covered by LNS (in a 600-second run).

LNS shows consistent but modest cluster coverage growth with increasing running time. In other words, on average, boundary candidates were found using LNS to cover one or two more clusters when increasing the execution time from 30 s to 10 min. In contrast, BCS shows more cluster coverage improvement over time, where five additional clusters were covered when searching for 10 min, both for BMI classification and Julia Date. It shows no such growth for bytecount or BMI. Still, it has also "saturated" for these SUTs already after 30 s, *i.e.,* it has covered the total number of clusters found after 30 s and thus has little to no potential for further improvements.

In many cases, BCS and LNS cover the same clusters, but some exceptions exist. For instance, only BCS found boundaries between the valid-error and error-error partitions

**Table 7 Statistics over the potential boundary candidate clusters covered by BCS and LNS.** We also show the number of clusters uniquely covered by each approach for each execution time setting. The Total Clusters column lists the number of clusters found by the summarisation method when run on all candidates found by any method in any run.

| | | Total | 30 seconds | | 600 seconds | |
| --- | --- | --- | --- | --- | --- | --- |
| **SUT** | **Strategy** | **Clusters** | **# Found** | **# Unique** | **# Found** | **# Unique** |
| bytecount | LNS | 8 | $5.25 \pm 0.6$ | 0 | $6.0 \pm 0.0$ | 0 |
| | BCS | | $8.0 \pm 0.0$ | 2 | $8.0 \pm 0.0$ | 2 |
| BMI-class | LNS | 15 | $14.95 \pm 0.2$ | 3 | $15.0 \pm 0.0$ | 0 |
| | BCS | | $9.35 \pm 0.9$ | 0 | $14.25 \pm 0.6$ | 0 |
| BMI | LNS | 6 | $6.0 \pm 0.0$ | 0 | $6.0 \pm 0.0$ | 0 |
| | BCS | | $6.0 \pm 0.0$ | 0 | $6.0 \pm 0.0$ | 0 |
| Julia Date | LNS | 11 | $4.9 \pm 0.3$ | 1 | $5.0 \pm 0.0$ | 0 |
| | BCS | | $2.7 \pm 1.1$ | 1 | $7.5 \pm 0.8$ | 6 |

for bytecount-clusters 7 and 8 (see 'Bytecount'). In contrast, considering the 30-second search, only LNS identified candidates in BMI classification that cover the transitions between (underweight, normal) and (normal, overweight)-clusters 8 and 13 (see Section 'BMI classification'). However, with increased execution time, BCS was the only strategy to find unique clusters (final column of Table 7). Particularly, if we look at Julia Date (10 min), BCS covers six unique clusters-including two clusters with "valid" outputs but unexpectedly long month strings. In RQ3, we further explain and compare these clusters for each SUT and argue their importance.

The comparisons above highlight the trade-off between time and effectiveness of the exploration strategies. Overall, LNS can be more effective in covering clusters in a short execution (BMI-class and Julia Data), but this is not always the case (*e.g.*, bytecount). And with more execution time, BCS generally catches up to LNS (BMI-class) and often surpasses it (bytecount and Julia Date) both on average and in the number of uncovered unique clusters. Clearly, attributes of the SUTs will affect cost-effectiveness, *e.g.*, the number of arguments in the input, (complexity of) specification, or the theoretical number of clusters that could be obtained to capture boundary behaviour.

From Table 7, we also note that the standard deviations are typically low, so the method is overall robust to random variations during the search. Still, we do note that the best method for Julia Date (BCS) only finds 7–8 of the total 11 clusters. This is not so for the other three SUTs, where it tends to find all of the clusters in a 600-second run.

For Investigation 2, Table 8 details the success rates of the boundary candidate summarisation. Summarisation is considered *failed* for a validity group where no boundary candidates could be found in that group. For instance, for 85% of the SUTs, no VE boundaries were found; thus, no summarisation could be attempted. The separation into good and bad summarisations is entirely based on the Silhouette score, where $SS \geq 0.6$ is considered good, otherwise bad. While there are differences between the exported and non-exported SUTs, nothing sticks out, except that the exported functions have a lower overall summarisation success rate, with 61% of the SUTs, compared to the 68% of the

**Table 8 Overall success statistics of the boundary candidate detection for Julia Base in Investigation 2 separated into two categories.** Successful detection with bad clustering quality ($SS < 0.6$) and good clustering ($SS \geq 0.6$).

| Exported | VE | | VV | | EE | | Total | |
|---|---|---|---|---|---|---|---|---|
| | bad | good | bad | good | bad | good | good | SUTs |
| yes | 14 (4%) | 38 (11%) | 37 (10%) | 109 (31%) | 13 (4%) | 127 (36%) | 216 (61%) | 354 |
| no | 12 (5%) | 29 (11%) | 34 (13%) | 119 (46%) | 1 (0%) | 75 (29%) | 175 (68%) | 259 |
| all | 26 (4%) | 67 (11%) | 71 (12%) | 228 (37%) | 14 (2%) | 202 (33%) | 391 (64%) | 613 |

**Table 9 The impact of the exploration strategy on the number of clusters identified during the 30-second runs.**

| Exported | Ground truth size | Strategy | Found | Unique |
|---|---|---|---|---|
| yes | 3.97 ± 4.65 | bcs | 3.36 ± 0.17 | 0.16 ± 0.69 |
| | | lns | 3.5 ± 0.13 | 0.1 ± 0.47 |
| no | 3.78 ± 5.06 | bcs | 3.02 ± 0.22 | 0.2 ± 0.68 |
| | | lns | 3.17 ± 0.19 | 0.16 ± 0.92 |

non-exported ones. For all Validity Groups, good clusterings far outnumber the bad ones. It should be noted that the total numbers for good clustering in Table 8 counts those SUTs for which good quality clustering could be obtained for *at least one* validity group - a number smaller than the sum over all high-quality clusterings over all validity groups. Given that candidates could be detected for roughly 70% of the SUTs (see RQ1), and 64% could successfully be summarised, summarisation seems successful in separating the candidates in most cases.

Table 9 looks at the exploration strategies that impact boundary detection by validating how many clusters could be covered per 30 s run (average and standard deviation). The finding capabilities are relatively stable for both strategies, but the simpler LNS detects most candidates for these short runs over the board.

> **Box 2.** *Key findings (RQ2):*
>
> The identified boundary candidates cover a diverse range of boundary behaviours. While local neighbour sampling (LNS) can often find more clusters in a short (30-second) run, boundary crossing search (BCS) catches up and often finds both more diverse and more unique candidates in longer runs. The framework is robust to random variations during the search and the number of unique behaviours found is mainly a function of the execution time and the characteristics of the SUT.

## RQ3-Identifying relevant boundaries

While the main focus here will be on Investigation 1, one subsection is devoted to the findings of the broader Investigation 2 and the generalizability of AutoBVA.

None of the SUTs in Investigation 1 have a formal specification to which we can compare the actual behaviour of the implementations as highlighted by the identified boundary

candidates. Thus, we cannot judge if any of the identified boundary candidate pairs indicate real faults.

Also, in practice, even if there was a formal specification, it might not be entirely correct or incomplete, *i.e.,* there might be situations/inputs for which it does not fully specify the expected behaviour. Human judgment would then be needed to decide what, if anything, would need to be updated or changed in response to unexpected behaviour uncovered during testing. In industry, it is more common with informal specifications consisting of requirements in natural language, which can further exacerbate these problems. However, a relative benefit of automated black-box boundary exploration with the techniques proposed here is that they can potentially help identify several of these problems in the specification (incompleteness or even incorrectness), the implementation (bugs), and/or both. And even if no issues are identified, our proposed techniques can help strengthen the test suite.

Below, we go through each SUT, in turn, and manually analyse the boundary pairs identified and if they actually did uncover relevant (expected and unexpected) behaviour or even indicate actual faults. We used the clusters identified by the summarisation process (see RQ2 above) as the starting point. For clusters of a size smaller than 50 boundary pairs, we went through all of them. For larger clusters, we randomly sampled pairs, stratified by the total size of the outputs and analysed them, from smaller sub-groups to larger ones until saturation, *i.e.,* looking at least at 50 pairs and going on further until no new, interesting or unexpected behaviour was found. For additional detail, we also calculated program derivative values using the Overlap Coefficient (based on 2-grams) function as output distance and checked all top-10 ranked pairs, per cluster. In the following, we highlight the critical findings per cluster and SUT.

To support our reporting on the manual analysis, we extracted tables with cluster representatives (see Tables 10–13). Unfortunately, some table entries had to be shortened for brevity. The original values and details can be found as part of the reproduction package (https://doi.org/10.5281/zenodo.7677012). Since the answers to RQ1 and RQ2 above indicated that BCS was sometimes more effective (even if not always as efficient as LNS), the tables have a column showing how many of the total candidates per cluster were found by BCS.

### Bytecount

Table 10 contains the representatives for the clusters identified for bytecount. All six members of cluster 4 for bytecount are the very natural and expected boundaries where the output string *suffix* changes from a lower value to the next, *e.g.,* from the smallest input pair of the cluster (999, 1000) with outputs ("999B", "1.0 kB") to the largest pair (999949999999999999, 999950000000000000) with outputs ("999.9 PB", "1.0 EB"). While the behaviour is not unexpected, it is essential that also these expected boundaries are identified. A tester can then more easily verify that the implementation corresponds to what is expected.

The six members of cluster 3 have a similar pattern to the ones in cluster 4, but here the *transition is within* each output string suffix category for the transitions from 9.9 to

**Table 10  Examples of representative candidates for each cluster, and corresponding validity group (Gr.) for bytecount.** We include the cluster coverage for BCS (BCS$_{\text{clust-cov}}$) as absolute values and percentage in percentage. Rows marked with an asterisk indicate clusters that are uniquely covered by BCS in a 600 s search. Some input values and exception types (BError refers to BoundsError) were abbreviated for brevity.

| ID | Gr. | Input 1 | Output 1 | Input 2 | Output 2 | BCS$_{\text{clust-cov}}$ |
|---|---|---|---|---|---|---|
| 1 | VV | -1 | -1B | 0 | 0B | 3 (100%) |
| 2 | VV | -10 | -10B | -9 | -9B | 34 (100%) |
| 3 | VV | 9,950 | 9.9 kB | 9,951 | 10.0 kB | 6 (100%) |
| 4 | VV | 999 | 999B | 1,000 | 1.0 kB | 6 (100%) |
| 5 | VV | 99,949 | 99.9 kB | 99,950 | 100.0 kB | 7 (100%) |
| 6 | VV | false | falseB | true | trueB | 1 (100%) |
| 7* | VE | 99...56 | 1000.0 EB | 99...57 | B.Error("kMGTPE", 7) | 1 (100%) |
| 8* | EE | 99...16 | B.Error("kMGTPE", 9) | 99...17 | B.Error("kMGTPE", 10) | 1 (100%) |

**Table 11  Examples of representative candidates for each cluster, and corresponding validity group (Gr.), for Julia Date.** We include the cluster coverage for BCS (BCS$_{\text{clust-cov}}$) as absolute values and percentages in parenthesis. Rows marked with an asterisk indicate clusters uniquely covered by BCS in a 600-second search. Some input values and exception messages were abbreviated for brevity. 'Err' refers to Errors in Julia due to months (Mon) or days out of range (oor).

| ID | Gr. | Input 1 | Output 1 | Input 2 | Output 2 | BCS$_{\text{clust-cov}}$ |
|---|---|---|---|---|---|---|
| 1* | VV | (-10000,2,3) | -10000-02-03 | (-9999,2,3) | -9999-02-03 | 8 (100%) |
| 2 | VV | (-1,9,3) | -0001-09-03 | (0,9,3) | 0000-09-03 | 38 (33%) |
| 3* | VV | (9999,5,9) | 9999-05-09 | (10000,5,9) | 10000-05-09 | 13 (92%) |
| 4* | VV | (75...81,2,21) | 25...50-60...91-02 | (75...82,2,21) | -25...50-12...77-30 | 1 (100%) |
| 5* | VV | (16...92,3,22) | 99...99-18...68-20 | (16...93,3,22) | 10...00-18...68-20 | 5 (100%) |
| 6 | VE | (0,2,0) | Err("Day: 0 oor (1:29)") | (0,2,1) | 0000-02-01 | 1560 (14%) |
| 7 | VE | (330,5,0) | Err("Day: 0 oor (1:31)") | (330,5,1) | 0330-05-01 | 111 (12%) |
| 8 | EE | (-8,3,-1) | Err("Day: -1 oor (1:31)") | (-8,3,0) | Err("Day: 0 oor (1:31)") | 6373 (18%) |
| 9 | EE | (0,0,92) | Err("Mon: 0 oor (1:12)") | (0,1,92) | Err("Day: 92 oor (1:31)") | 108 (12%) |
| 10* | EE | (0,4,99) | Err("Day: 99 oor (1:30)") | (0,4,100) | Err("Day: 100 oor (1:30)") | 7 (87%) |
| 11* | EE | (0,9...9,0) | Err("Mon: 9...9 oor (1:12)") | (0,1...0,0) | Err("Mon: 1...0 oor (1:12)") | 3 (100%) |

10.0, *e.g.*, the input pair (99949999, 99950000) with outputs ("99.9 MB", "100.0 MB"). Since the outputs in such pairs differ in length, our output distance function detects them. Cluster 5 has the same six transitions but between 99.9 and 100.0 and one extra boundary pair for the exabyte suffix ("EB"). Since this is the last suffix class and thus does not switch over to the following suffix at the value of "1000.0 EB". Since it is not apparent what the behaviour at "1000.0 EB" should be, not all specifications might cover it, and thus it would be important to test and check against expectations.

Cluster 1 contains three candidates all *within* the "B" byte suffix group, covering the transitions from "-1B" to "0B", from "9B" to "10B", and from "99B" to "100B". While the transition from zero to negative one seems like a natural boundary, one could argue that the other two boundaries are less fundamental and are an artefact of our specific choice of output distance function (string distance, here detecting the difference in lengths between "9" and "10" *etc.*). But the extra cost for a tester to verify they are there seems

**Table 12  Examples of representative candidates for each cluster, and corresponding validity group (Gr.), for BMI classification.** We include the cluster coverage for BCS ($BCS_{clust-cov}$) as absolute values and percentages in parentheses. Some input values and exception messages were abbreviated for brevity. 'DomErr' refers to DomainError.

| ID | Gr. | Input 1 | Output 1 | Input 2 | Output 2 | $BCS_{clust-cov}$ |
|---|---|---|---|---|---|---|
| 1 | VV | (1,0) | Underweight | (1,1) | Severely obese | 19 (100%) |
| 2 | VV | (21,1) | Normal | (21,2) | Severely obese | 5 (100%) |
| 3 | VV | (26,1) | Underweight | (26,2) | Obese | 5 (100%) |
| 4 | VV | (29,1) | Underweight | (29,2) | Overweight | 3 (100%) |
| 5 | VV | (29,2) | Overweight | (29,3) | Severely obese | 2 (100%) |
| 6 | VV | (105,20) | Underweight | (105,21) | Normal | 115 (68%) |
| 7 | VV | (102,31) | Obese | (102,32) | Severely obese | 79 (82%) |
| 8 | VV | (100,22) | Normal | (100,23) | Overweight | 32 (36%) |
| 9 | VV | (51,6) | Overweight | (51,7) | Obese | 39 (41%) |
| 10 | VV | (102,26) | Obese | (103,26) | Overweight | 150 (37%) |
| 11 | VV | (110,28) | Overweight | (111,28) | Normal | 183 (44%) |
| 12 | VV | (1,002,3,012) | Severely obese | (1,003,3,012) | Obese | 2,766 (95%) |
| 13 | VV | (100,255,18,544,584) | Normal | (100,256,18,544,584) | Underweight | 3,012 (97%) |
| 14 | VE | (-1,0) | DomErr(...) | (0,0) | Severely obese | 7,457 (4%) |
| 15 | VE | (1,-1) | DomErr(...) | (1,0) | Underweight | 7,995 (4%) |

**Table 13  Examples of representative candidates for each cluster, and corresponding validity group (Gr.), for BMI.** We include the cluster coverage for BCS ($BCS_{clust-cov}$) as absolute values and percentages in parenthesis. The exception messages were abbreviated for brevity.

| ID | Gr. | Input 1 | Output 1 | Input 2 | Output 2 | $BCS_{clust-cov}$ |
|---|---|---|---|---|---|---|
| 1 | VV | (0,1) | Inf | (1,1) | 10000.0 | 4039 (1%) |
| 2 | VV | (90,8) | 9.9 | (90,9) | 11.1 | 6129 (5%) |
| 3 | VV | (100,10) | 10.0 | (101,10) | 9.8 | 4194 (3%) |
| 4 | VE | (-1,0) | DomainError("H or W negative...") | (0,0) | NaN | 2 (100%) |
| 5 | VE | (-1,1) | DomainError("H or W negative...") | (0,1) | Inf | 9101 (1%) |
| 6 | VE | (1,-1) | DomainError("H or W negative...") | (1,0) | 0.0 | 6277 (3%) |

slight. In the general case, there is, of course, a cost involved in having to screen very many candidate pairs. However, the transition from zero to negative inputs should prompt a tester to consider if this should really be allowed (in the specification) or not.

The 34 members of cluster 2 are of more questionable relevance as they are all the transitions from "-9B" to "-10B", "-99B" to "-100B", and so on up for every output string length between 2 up to 36. An argument can be made that it is suitable for a tester to check if *negative inputs* should even be allowed and, if so, how they should be handled. But having more than a few such examples is probably not adding extra insight, and the transition from 0 to -1 was already covered by the candidate in cluster 1 above.

The final valid-valid (VV) cluster (6) for bytecount contains the single pair (false, true) with outputs ("falseB", "trueB"). This comes from the fact that in Julia, the "Bool" type is a subtype of "Integer", and our tested Julia implementation of bytecount only specifies that inputs should be integers; booleans are thus generated during the search, and this pair

is found. Again, it is unclear if this input type should be allowed, but we argue that a tester must know of this implemented behaviour to decide if it is good enough or needs to be addressed. Even if one chooses to keep this functionality in the implementation, adding it as a test case to the test suite, at least as a kind of documentation, seems valuable.

There is a single valid-error (VE) cluster for bytecount (7) that has the single member (99999999999994822656, 99999999999994822657) where the first output is the string "1000.0EB" while the latter throws the exception *BoundsError("kMGTPE",7)*. The Julia exception indicates that the implementation tried to access the string "*kMGTPE*", of length 6, at position 7. Similarly, there is a single error-error (EE) cluster (8) where the exception thrown changes from *BoundsError("kMGTPE",9)* to *BoundsError("kMGTPE",10)*. Having found three inputs for which there are different kinds of BoundsErrors thrown, it is then obvious that there will be other such transitions, *i.e.,* between the errors accessing the string at position 7 and those at position 8, *etc*. Since our output distance only detects differences in length it does not identify the transition from 7 to 8 or 8 to 9 but picks up the transition from 9 to 10. This shows some of the trade-offs in the selection of the output distance function; while the one we have chosen here is very fast and does find a lot of relevant boundary pairs more fine-grained detection can be possible with more sensitive output distance functions.

### Julia Date

Cluster number 4 for the Julia Date SUT, shown in Table 11, contains a single boundary candidate pair which shows an unexpected switch in the outputs despite both being valid Dates. The pair also has among the largest program derivative values found overall (0.634). This candidate uses very large values for the year input parameter (757576862466481 and its successor), coupled with "normal" month and day values, but the outputs have no resemblance to the inputs and also switches the sign for the year in the date outputs (outcomes are 252522163911150-6028347736506391-02 and -252522163911150-12056695473012777-30, respectively). Even if such high values for the year parameter are not very likely in most use cases, we argue that it is still useful for a developer or tester to know about this unexpected behaviour. They can then decide if and how to handle the case, *i.e.,* update either the specification and the implementation or document the behaviour by adding a test case.

The pairs found in the valid-valid cluster 5 similarly all happen for large values of the year input parameter but differ from cluster 4 in that the output dates are more similar to each other and typically only differ in one of the Date elements, *e.g.*, year or month. Correspondingly the PD values are much lower (minor variation around 0.20).

Cluster 1 contains pairs where all years are negative and switches from one order of magnitude (all nines, *e.g.*, "-9999-02-03", to the next one followed by zeros, *e.g.*, "-10000-02-03"). Since the PD output distance is the output string length, many such boundaries (for a different number of nines) are found. While the outputs in this cluster have similarities to the ones in cluster 5 above, the latter does not have inputs that correspond to the outputs. For cluster 1, the input years correspond to the output years. Splitting these into two clusters thus makes sense.

The largest valid-valid cluster (2) contains many pairs that only differ in the month and day, while the year always goes from -1 to 0.

Clusters 6 and 7 have pairs where one input leads to an Error while the other leads to a valid Date. Both clusters have errors that complain about an invalid day or an invalid month. We could not identify an apparent reason why these two clusters were not merged. Most likely, it is just an artefact of the clustering method we used in the summarisation step.

The remaining clusters all raise exceptions for both inputs. Cluster 11 has only 3 pairs complaining about invalid month inputs, all of a month transition from a number of nines to the next order of magnitude. Similarly, the pairs of cluster 10 all complain about invalid day inputs, all being variations of nines and the next order of magnitude. This cluster is larger since there are more unique exceptions of this type. The error message depends on the month since different months have differing numbers of allowed day ranges. Cluster 9 then has pairs where one input leads to an argument error for an invalid month and the other for an invalid day. Cluster 8 then mainly has both inputs being invalid day values, although some combined pairs (both invalid month and invalid day) are also in this one.

### BMI classification

For the BMI classification SUT (Table 12), many clusters show the boundaries between "adjacent" output classes, *i.e.,* underweight to normal (clusters 6 and 13), normal to overweight (8 and 11), overweight to obese (9 and 10), and obese to severely obese (7 and 12). There are two clusters for each such boundary, and they differ only in the order of the outputs, *i.e.,* cluster 6 has the underweight output first while cluster 13 has normal first, *etc.* We can also note that these clusters are relatively large, with the smallest one (cluster 10) containing 93 pairs up to the largest one (cluster 13) containing 871 pairs. This is natural since the formula for calculating BMI allows many different actual inputs to be right on the border between two adjacent output classes.

In contrast to the "natural" boundaries above, clusters 1 to 5 all contain fewer boundary candidates (from 2 to 19), but all correspond to transitions between output classes that are unexpected. For example, cluster 1 contains extreme examples of inputs that are very close but where the output class jumps from underweight to severely obese. We note that all of these clusters happen for very extreme input values, and it is likely that we can address many of these problems by putting limits on the valid ranges of each of the inputs. However, it is essential that our method was able to find a transition not only between some of these non-adjacent output classes but for several combinations of them.

Finally, the method identified a large number of valid-error pairs at either end of the output class adjacency scale. Cluster 14 has pairs that go from severely obese to an input domain error where one of the values is negative, while cluster 15 has pairs that go from the underweight output class to input domain errors. The sizes of these clusters are very large and it is not likely a tester would get much extra benefit from having so many candidate pairs. Future work can thus explore ways of focusing the search to avoid finding many candidate pairs in the same cluster.

**Table 14 The top three lists of SUTs in Julia Base from Investigation 2 with the highest clustering Silhouette scores in VV, VE, and EE, respectively.** The BCS coverage is the number of boundary candidate clusters for the 30 s runs ($\mu \pm \sigma$). In bold font are the SUTs we describe in more detail in the article.

| SUT | Silhouette Score | | | # Clusters | | | | # BCS coverage per run |
|---|---|---|---|---|---|---|---|---|
| | VV | VE | EE | VV | VE | EE | Total | |
| power_by_squaring(2) | 0.92 | 0.93 | 0.63 | 15 | 10 | 16 | 41 | $29.9 \pm 2.13$ |
| tailjoin(2) | 0.91 | 0.69 | 0.9 | 3 | 14 | 8 | 25 | $17.9 \pm 2.33$ |
| **max(2)** | 0.90 | 0.82 | 0.92 | 4 | 3 | 3 | 10 | $9.0 \pm 0.82$ |
| **cld(2)** | 0.79 | 0.97 | 0.0 | 2 | 4 | 0 | 6 | $6.0 \pm 0.0$ |
| fldmod1(2) | 0.83 | 0.96 | 0.88 | 2 | 9 | 2 | 13 | $11.7 \pm 0.82$ |
| fld(2) | 0.67 | 0.96 | 0.0 | 2 | 4 | 0 | 6 | $6.0 \pm 0.0$ |
| **max(2)** | 0.9 | 0.82 | 0.92 | 4 | 3 | 3 | 10 | $9.0 \pm 0.82$ |
| tailjoin(2) | 0.91 | 0.69 | 0.9 | 3 | 14 | 8 | 25 | $17.9 \pm 2.33$ |
| fldmod1(2) | 0.83 | 0.96 | 0.88 | 2 | 9 | 2 | 13 | $11.7 \pm 0.82$ |

### BMI

For the BMI SUT (Table 13), there are only six clusters identified, with clusters 4 to 6 all having one input that leads to a domain error raised while the other output is either NaN (cluster 4), Infinity (5), or 0.0 (6). While cluster 4 is rare and only happens for two specific input pairs, the other clusters are huge. This reflects the fact that there are many ways to create an infinite output (height of zero and weight is any value) or zero output (weight is zero and height is any value). Even though the clustering for the VE group was of objectively low quality (see Table 6), clusters 4–6 clearly highlight different kinds of boundaries. This suggests that high-quality clustering is not a prerequisite for good boundary summarisation. It can be expected to be a heuristic and imperfect process as it is very generic. As long as only a few interesting candidates can be differentiated and selected, as was possible here (with a poor Silhouette Score of 0), the results may be helpful for a tester. How this generalizes shall be investigated in future research.

The valid-valid cluster 1 has pairs where the first outputs infinity while the second input leads to a normal, floating point output. The largest cluster is the valid-valid cluster 2 which has pairs with normal, floating point outputs that differ only in their length. Of course, there are many such transitions, and our system identifies many of them, but it is not clear that a tester would be helped by some of them more than others. Sorting just by length and including a few such transitions will likely be enough.

### RQ3 and Investigation 2

To supplement the quantitative findings for RQ1 and RQ2, we manually analysed some of the SUTs from Investigation 2. As each of the three validity groups, namely VV, VE, and EE, produces a unique Silhouette score, we have selected three SUTs with the highest score per group and present them in Table 14. Our analysis reveals that the clusters identified by AutoBVA correspond to the expected behaviours/clusters in the documentation for most of the investigated SUTs, with a few exceptions. To illustrate our analysis, we have chosen

**Table 15  Examples of representative candidates for each cluster, and corresponding validity group (Gr.), for cld(2).** We include the cluster coverage for BCS ($\text{BCS}_{\text{clust-cov}}$) as absolute values and percentages in parentheses.

| ID | Gr. | Input 1 | Output 1 | Input 2 | Output 2 | $\text{BCS}_{\text{clust-cov}}$ |
|----|-----|---------|----------|---------|----------|-------------|
| 1 | VV | (-12,-2) | 6 | (-12,-1) | 12 | 6,016 (61%) |
| 2 | VV | (-10,10) | 1 | (-9,10) | 0 | 3,532 (67%) |
| 3 | VE | (-644...482,false) | DivideError() | (-644...482,true) | -644...482 | 1,995 (63%) |
| 4 | VE | (-11,-1) | 11 | (-11,0) | DivideError() | 1,310 (65%) |
| 5 | VE | (-2,-1) | 2 | (-2,0) | DivideError() | 6,510 (62%) |
| 6 | VE | (339...990,-1) | 917...466 | (339...990,0) | DivideError() | 524 (71%) |

two SUTs from the table, namely, `cld(2)` (which had the highest VE Silhouette score) and `max(2)` (which had the highest EE and 3rd highest VV Silhouette score).

Looking into the SUTs for which no useful summarisation was obtained, some common properties could be observed. A larger number of these functions output memory address information, particularly in the error messages when they throw exceptions. This makes them non-deterministic and, therefore, harder to summarise as the memory information adds noise to the feature extraction for the clustering.

Table 15 shows the six (6) clusters identified for the `cld(2)` SUT in one of the AutoBVA runs. This function is documented as a special case of the more general `div(2)` function for integer division but always rounding up to the nearest, larger integer divisor. The three VE clusters, 4, 5, and 6, all represent the boundary between a valid input and one that leads to a division by zero, which raises an exception. Thus, All these candidates are expected for any function implementing division.[14] The candidates in (VE) cluster 3 are more unexpected. They also lead to a division by zero exception but result from dividing by the boolean value false. While this is consistent with Julia's type system, where the boolean type is a subtype of the integer type hierarchy, it isn't certain if a developer or tester would realize this nor what the correct behaviour should be. Thus it is useful that AutoBVA identifies the cluster. The two VV clusters, 1 and 2, are also meaningful and represent the boundary between a negative and positive output (cluster 2) or outputs of different lengths (cluster 1).

Table 16 shows the ten (10) clusters identified for the `bytecount(2)` SUT on one of the AutoBVA runs. This function should return the maximum value of the arguments supplied to it. The VV clusters identified are the expected ones with different output length boundaries for both positive (cluster 3) and negative (cluster 1 and 2) arguments, as well as a boundary for the cases when arguments are boolean (cluster 4). The three VE clusters (5–7) are unexpected and, at first sight, hard to understand. In cases where one argument is −1, and the other argument is a very large positive number requiring either 64 (cluster 5 and 7) or 128 (cluster 6) bits to represent, the SUT throws an exception. A similar problem can be seen for the EE clusters 8–10, but here the exception is thrown on both sides of the boundary resulting in different lengths of the outputs.

We investigated these unexpected boundary candidates for `max(2)` further and argued that this is either a genuine bug or, if not, should be documented so that developers are aware of it. The sample Julia REPL interactions in the Listing 1 highlights the problem: for

[14]We could not identify a semantically meaningful difference between the three clusters; the method splits the candidates up into clusters based on the length of the actual inputs

**Table 16 Examples of representative candidates for each cluster, and corresponding validity group (Gr.), for max(2).** We include the cluster coverage for BCS ($BCS_{clust\text{-}cov}$) as absolute values and percentages in parentheses. For brevity, we abbreviate some values and Errors (InexErr refers to the InexactError type).

| ID | Validity group | Input 1 | Output 1 | Input 2 | Output 2 | $BCS_{clust\text{-}cov}$ |
|----|----------------|---------|----------|---------|----------|--------------------------|
| 1 | VV | (-1,-3) | -1 | (0,-3) | 0 | 10 (7%) |
| 2 | VV | (-10,-16) | -10 | (-9,-16) | -9 | 237 (48%) |
| 3 | VV | (3,9) | 9 | (3,10) | 10 | 820 (60%) |
| 4 | VV | (false,false) | false | (true,false) | true | 2 (100%) |
| 5 | VE | (-1,229...193) | InexErr(..., UInt64, -1) | (0,229...193) | 229...193 | 60 (60%) |
| 6 | VE | (-1,455...560) | InexErr(..., UInt128, -1) | (0,455...560) | 455...560 | 63 (55%) |
| 7 | VE | (195...517,-1) | InexErr(..., UInt64, -1) | (195...517,0) | 195...517 | 6 (42%) |
| 8 | EE | (-100,181...675) | InexErr(..., UInt64, -100) | (-99,181...675) | InexErr(..., UInt64, -99) | 91 (64%) |
| 9 | EE | (187...829,-100) | InexErr(..., UInt128, -100) | (187...829,-99) | InexErr(..., UInt128, -99) | 83 (54%) |
| 10 | EE | (568...833,-10) | InexErr(..., UInt64, -10) | (568...833,-9) | InexErr(..., UInt64, -9) | 43 (26%) |

unsigned integers of sizes 8, 16, or 32 bits max(2) correctly returns the largest argument, even when the other argument is negative. But this is not so for 64 or 128-bit unsigned integers, where an exception is thrown. It seems that if the two arguments have different types the implementation coerces the smaller one into the type of the larger one. Thus, if the larger argument is an unsigned integer type (of size 64 or 128) an exception is thrown since the other argument, here $-1$, cannot be coerced into a value of the large unsigned type.

### Summary for RQ3

Taken together, our manual analysis of the identified clusters and their boundary candidates shows that the method we propose can reveal both expected and unexpected behaviour for each of the tested SUTs. Using bytecount as an example, 21 expected boundaries were automatically identified, and divided into three main groups:

**Listing 1** Bug identified when running AutoBVA on the Julia Base function max(2). If one argument is negative and the other one is an unsigned integer of 64 (or 128) bits an exception is thrown. This does not happen for smaller unsigned integers.

```
1  julia> max(-1, UInt8(0))
2  0
3
4  julia> max(-1, UInt16(0))
5  0
6
7  julia> max(-1, UInt32(0))
8  0
9
10 julia> max(-1, UInt64(0))
11 ERROR: InexactError: check_top_bit(UInt64, -1)
```

1. transitions between consecutive byte suffix partitions, *e.g.*, "999.9 kB" to "1.0 MB" (6 candidates),
2. transitions from zero to negative values, "0B" to "-1B" (1),
3. transitions within the same byte suffix partitions, *e.g.*, "9B" to "10B", "9.9 MB" to "10.0 MB", and "99.9 GB" to "100.0 GB" (14).

Of these, we argue that the first two groups (1 and 2) are expected and natural, while a tester can decide if and, if so, how many from group 3 to include in the test suite. The method also identified four (4) boundaries for bytecount that we argue were unexpected:[15]

1. transition from "999.9 EB" to "1000.0 EB",
2. transition between boolean inputs "falseB" to "trueB",
3. transition from the valid output, "1000.0 EB" for input 999999999999994822656, to an exception, `BoundsError("kMGTPE", 7)` for input 999999999999994822657,
4. transition from two different exceptions, `BoundsError("kMGTPE", 9)` for input 9999999999999905201 04160854016 and `BoundsError("kMGTPE", 10)` for input 999999999999990520 104160854017.

In hindsight, a tester can likely understand the reasons for these boundaries. Still, we argue that it is not apparent from just looking at the implementation or specification that these boundaries exist. Even though the very simple output distance function we have used cannot detect the additional error-error boundaries, we can understand to be there (between bounds error 7 and 9, for example), it would be relatively simple to find them with a more focused search once we know to look for them. This also points to future work investigating alternative output distance functions or even hybrid search approaches that apply multiple distance functions for different purposes.

For bytecount, another 34 boundary candidates were identified that were also unexpected, but where we judge, it is less likely that a tester would include them all in a test suite. These are the transitions between different sizes of negative inputs, *e.g.*, from "-9B" to "-10B" and so on. A tester might want to sample some of them to ensure proper treatment by a refined implementation. Still, since the transition from zero to negative one has already been found, the additional value is relatively limited.

For the Julia Date SUT, several clusters were of more debatable value. In particular, in the error-error group, likely, a tester would only have selected some typical examples from the identified clusters. While there were some differences between the clusters, they essentially just differed in whether the month or day inputs led to an exception being thrown. We note that the boundary transitions for invalid day inputs covered all the different months (30 to 31 valid-error transition for June, 31 to 32 for August, 28 to 29 for February, *etc.*). However, the clustering was insufficient to separate them into individual groups, making the manual screening less efficient.

For BMI-class, most clusters contained relevant boundaries. While all the expected boundaries between consecutive, ordinal outputs (like normal to obese) were identified, the method also identified many unexpected boundaries between non-consecutive output categories.

With its numerical outputs, a much larger number of candidates were identified for BMI. Still, the summarisation method successfully distilled them to only five clusters, making it relatively easy to screen them manually. Even if expected and relevant boundaries were found, it is harder, in this case, to define and judge if other boundaries should be found in the large, valid-valid groups. In this case, it was unclear that the output distance function used was fine-grained enough to pick out essential differences.

[15] Since the example number below is very long, the numbers might be split between different lines.

While all eight clusters for bytecount and BMI contained at least one boundary candidate that we argue a tester would like to look at, this was not the case for all the other SUTs. For example, for BMI-class, several clusters differed only in the order of the outputs. This should be refined in future work on the method. There were also cases where clusters seemed unnecessarily split into multiple clusters for which we could not discern any apparent pattern or semantic reason. This is most likely an artefact of the clustering method and the features we used as input. Still, we argue that since the number of clusters identified was relatively limited, it would not be a significant cost for testers to screen them all.

While the number of identified candidates for bytecount was low (58), the clustering for summarisation helped identify attractive boundary candidates. This was even more evident for the more complex SUTs where the number of candidates identified was huge; summarisation is thus necessary, and clustering is one helpful way to achieve it. Future work should investigate how to refine the summarisation further to reduce the number of candidates a tester has to look at.

The above findings were also seen for the larger number of SUTs sampled from Julia's Base library for Investigation 2. Both expected and unexpected boundaries were identified, and for the latter, the boundary highlighted ''buggy'' behaviour. Still, the number of clusters can often be large, and in several cases, the only difference between clusters we could identify was the length differences in their output. While this is a direct consequence of the chosen output distance function, it does lead to more manual work for the tester. Future work should thus investigate other complementary ways to summarise the boundary candidates that AutoBVA identifies.

> **Box 3.** *Key findings (RQ3)*:
>
> The AutoBVA method can successfully identify expected and unexpected boundaries without using a specification or white-box information. While the value of identified candidates ultimately depends on the tester, the summarisation via clustering helped focus the manual screening. The approach identified boundaries for some programs that indicate implementation, specification, or documentation bugs. Further refinement to the summarisation method should be investigated to minimize the number of different clusters a tester must manually inspect.

## DISCUSSION

In this section, we provide a more general discussion on lessons learnt and future work, followed by dedicated sections on tester actions for various situations ('Boundary candidates types and tester actions') and the threats to validity ('Threats to validity').

Our results show that relevant boundaries in the input space and behaviour of programs can be identified automatically without needing a specification, white-box analysis, or instrumentation. This is important since it can help make boundary value analysis and testing more automated and systematic. While these techniques for quality assurance and testing have been advocated for a long and sometimes even been required by standards

and certification bodies, prior work has relied, for effective results, on the creativity and experience of the tester performing them.

By building on the vision from *Feldt & Dobslaw (2019)*, *Dobslaw, de Oliveira Neto & Feldt (2020)* and coupling their proposal to simple search algorithms, the system we propose here enables augmenting the testers performing boundary value analysis and testing by automatically identifying and presenting them with candidate input pairs that are more likely to reveal actual boundaries.

Our experimental validation shows that for the investigated SUTs, the system could identify many boundary candidate pairs that also cover a diverse range of behaviours. These results were also robust over multiple executions, despite the stochastic nature of the algorithms used. Manual screening showed that many relevant (important) boundary candidates were automatically identified, both those that could be expected for the investigated SUTs and relevant but unexpected ones that we argue would have been harder for a tester to think of.

We investigated two different search strategies within our overall framework. The simpler one, local neighbour search (LNS), is more directly similar to random testing (in automated testing) but with a local exploration around a randomly sampled point. It identified more boundary candidates but, even if given a longer execution time, did not find as diverse types of candidates as the other strategy. The latter, boundary crossing search (BCS), was tailored specifically to the problem at hand by first identifying inputs in two different input partitions and then "honing" in on the/any boundary between them. BCS needs more computational resources but consistently finds as many or more diverse clusters of candidates than LNS. Both strategies performed notably better when sampling broadly and bit-uniformly over the entire input space - this was, in fact, critical for LNS to identify boundaries overall (see Appendix A).

Regardless of the search strategy used, for our system to be beneficial to actual testers, a critical step is how to group and summarise the set of candidates found. While this is not the main focus of this study, we show that basing the grouping on the type of outputs of the pair and then clustering them based on their within- and between-pair distances can be helpful. However, our experiments also uncovered challenges in this approach that should be investigated in future work, *i.e.,* how to avoid showing too many groups (clusters) as well as candidates to a tester.

The key idea that our system builds upon is the program derivative, a way to quantify the rate of change for any program around one of its inputs. Our essential choice in this study was to use a fast but exceedingly simple distance function for outputs. By simply comparing the length of the outputs in a pair, we can only detect a difference that leads to differing lengths after stringifying them. This will not always be enough, for example, for functions where all outputs are the same length despite being different. Given this major limitation in our experiments, our results are encouraging; we can find relevant boundaries despite this simplification. One reason is likely that if outputs differ in some way, they often will also vary in their length. Another reason can be that by using such a fast but coarse-grained distance function, we can explore larger parts of the search space. Even the reasonably simple Overlap Coefficient string distance function, which would detect more

fine-grained differences in outputs, would be at least an order of magnitude and possibly more, slower. And more advanced methods like the compression distances would be orders of magnitude slower yet. Future work should examine the trade-off between fast but coarse and slower but more fine-grained distance functions. We note that the system need not select only one distance function; hybrid solutions could be tried to get a coarser view of the boundaries and then zooms in for further details in different sub-partitions.

We also saw limitations in the clustering approach we used in the summarisation step. In several cases, our manual analysis of the clusters it produced did not uncover any clear semantic difference between clusters. Instead, they mainly seemed to differ in the length of their inputs or outputs. Future work should investigate alternative summarisation methods, in particular clustering ones. Since the summarisation is done offline, outside of the search process, there is potentially more time for this analysis step, so more complex, costly and fine-grained distance functions are likely relatively more helpful.

While our results constitute evidence that the proposed approach has value, it is methodologically challenging to judge how complete a set of boundaries the method can find. There is no clear baseline to compare to, given that boundary value analysis has traditionally been a manual activity heavily dependent on a clear and detailed specification and the tester's experience. Still, future work should perform user studies, both "competing" with testers doing purely manual analysis and "in tandem" with testers to evaluate the added value of the automated system. And while this is a first step towards an automatic extraction of boundary pairs and, in extension, the creation of test cases, we acknowledge that entirely relying on *bottom-up* test case creation from the implementation only is not going to be sufficient in the general case. Using specifications as input into the process would help keep intent and actual behaviour aligned.

The choice of distance function will also be critical in future work that evaluates the approach on software with inputs and/or outputs of complex, non-numeric types. Here we have focused on and experimented with many SUTs with a small number of integers as inputs using Euclidean distance. It is natural for such inputs to use the absolute difference between inputs as the distance function. However, if the inputs are two complex trees or a directed acyclic graph and a vector of complex objects, it will be less clear which distance function to choose or the relative benefits of different choices. Using multiple distance functions to reveal a diverse range of boundary candidates could be considered.

Similarly, more complex mutation operators will be needed for complex types since there is no obvious, or at least not a single, way to mutate a complex object by a small amount. While our empirical evaluation does not show that the approach can be applied to all types of software, we consider the choice of mutation operators and distance functions to be relatively orthogonal to the overall approach used. Any automated testing approach that needs to execute SUTs with inputs of a complex type will need to grapple with these issues, *e.g.*, how to generate and mutate relevant inputs. This also means that there is a plethora of known techniques and tools that can be leveraged in this. For example, search-based techniques (*McMinn, 2004*; *Afzal, Torkar & Feldt, 2009*) can be used and have been investigated explicitly for complex test data generation (*Feldt & Poulding, 2013*). There is also active work on grammar-based data generation and fuzzing (*Havrikov &*

**Table 17  Different actions a tester can take given one or a set of related automatically identified boundary candidates and which artefacts will likely change for each action.**

| Action | Specification | Implementation | Test suite |
|---|---|---|---|
| Skip | — | — | — |
| ExtendTests | — | — | X |
| Debug | — | X | X |
| Clarify | X | — | (X) |
| Refine | X | X | X |

*Zeller, 2019*; *Eberlein et al., 2020*) and creating input grammars or specifications from concrete examples or executions (*Bastani et al., 2017*; *Steinhöfel & Zeller, 2022*). Future work should explore which techniques to leverage when evaluating our approach to SUTs with more complex, structured inputs.

While writing this article, we sought confirmation from the Julia core developers by stating an issue on the official Github Julia repository regarding the max(2) behaviour from Investigation 2 (https://github.com/JuliaLang/julia/issues/48971). They argued that this behaviour was not unexpected, but we then concluded that the documentation was unclear on this point. Consequently, we have submitted clarifying documentation pull request (https://github.com/JuliaLang/julia/pull/48973), which has since been merged into the main branch of the language (https://github.com/JuliaLang/julia/commit/e4c90e22e999e85268fc5465b2840df6f4f2fb94). Our manual analysis of boundary candidates indicated that different types of candidates exist and differ in the kind of action they are likely to lead to. The max (2) candidate, for instance, would require *clarification*, which we link to a need for updates in the specification/documentation and (potentially) extension of the test suite, but no implementation adjustments. In the next section, we give more examples and discuss these action types in more detail. Finally, we then conclude the discussion with limitations and threats to validity.

## Boundary candidates types and tester actions

Table 17 shows five actions a tester can typically take concerning one or a set of similar boundary candidates. The columns to the right show the artefacts likely to be affected in each case: the specification, the implementation, or the test suite.

A *Skip* action typically means that either the tester does not find the candidate(s) relevant/useful, or it is handled correctly by the implementation, clear enough in the specification, and tested well enough. Even in the latter case, it is crucial that an automated testing method can also identify boundaries of this type; if it frequently misses them, a tester might lose confidence in the tool. An *ExtendTests* action would happen when the identified behaviour is correct according to the specification but not yet well covered by the test suite. The *Debug* action occurs when a fault exists in the implementation, *i.e.*, the specification describes the correct behaviour, but the implementation does not comply. The *Clarify* situation is a consequence of a behaviour that is intended but not explicitly specified. Finally, the *Refine* action is a consequence of a boundary that is not desired but when the specification is either incorrect or incomplete. In this situation, we must clarify

the specification and debug/fix the implementation. For the last actions, the test suite will also typically be extended to ensure the same problem does not occur in the future.[16]

One type of boundary candidate we identified in the manual analysis was the *under-specified input range*, *i.e.,* for inputs that are not expected, which leads to unexpected behaviour. One example, for BMI-class, was the candidate with outputs that move from Underweight to Severely Obese by a change in body weight from 0 to 1 kg for a person with a height of 1 cm. Another example is Julia Date, with a very large year input that led to nonsensical outputs that showed no resemblance to the inputs, likely due to some overflow in the internal representation of date values. This type of boundary would typically lead to either a *Debug* or a *Refine* action, depending on whether an update of the specification is meaningful/necessary or not.

Other problems were identified concerning the `bytecount` SUT. We used this SUT since it is known as the most copied (Java) code snippet on StackOverflow and has several bugs as described in *Lundblad (2019)*: a rounding error, correctness for large values over 64 bits, and the fact that negative inputs are not appropriately handled. As described in our results above, the latter was clearly identified and can be seen as an *under-specified* problem. It likely leads to a Refine action since it is unclear how to handle negative inputs from the specification. The correctness for large values is still a problem even though the Integer type also includes 128-bit integers and the BigInt type to handle arbitrarily large integers in Julia. However, the problem manifests in the boundary candidates that lead to BoundsErrors since they also try to access larger byte suffix categories after exabytes ("EB"). A Refine or at least a Debug action is probably called for. The boundary candidates capturing the rounding error of `bytecount`, *e.g.,* kB to MB, were all identified in the searches (cluster 5 in Table 10) and represent *wrongly positioned boundaries*. While it is unclear if the rounding error can be directly spotted, even given the boundary examples in the cluster, it is contained in all the cluster boundaries. A diligent tester is likely to investigate critical boundaries in depth, and the fact that they are all in the same cluster can help make this checking more likely. This will lead to a Debug action.

### Threats to validity

Our analyses involve simple comparisons through descriptive statistics (RQ1 + RQ2) and qualitative analysis of the clusters (RQ3). We mitigate conclusion validity threats in our quantitative comparison by focusing on how the sampling strategies complement each other rather than simply comparing which is better.

By analyzing the uniqueness of candidates and clusters, we identified essential trade-offs. For RQ3, we did not exhaustively inspect all candidates in larger clusters. Hence there is a risk we miss important ones. We mitigate this risk by random sampling and iterative analysis until saturation in conclusions.

We operationalize the constructs of our proposed approach with a limited set of treatments and dependent variables. Consequently, the primary construct validity threats are our choices of (1) configuration (string length for PD, mutation operators, sampling methods, *etc.*), (2) SUTs with integer inputs, (3) measures of uniqueness, clustering, and coverage/inspection of clusters. However, given the novelty of the approach, we argue it

[16] For *Clarify*, there might, though, be situations where this is impractical, for instance, the max(2) SUT discussed above, which is why we set parentheses around the selection in Table 17.

would be very difficult to analyze the overall results if we used more advanced or just more choices for these contextual factors. Future work is needed to understand performance implications better as these factors vary.

We use clustering as an attempt for summarising boundary candidates into a presentable subset for testers.

To a certain degree, the "relevance/importance of a boundary" is subjective. Hence an optimal or perfect summarisation is hardly attainable. It should also be mentioned that in a real-world situation, a single search run with sufficiently many boundary candidates should suffice to do both the clustering and summary extraction - which will be faster and simpler than running many times over and building an overall clustering model over the entire set of found candidates. However, the number of clusters found in each run was smaller than the total number for some SUTs, even for BCS, so the effect of execution time needs further study in the future.

We mitigate internal validity threats by doing pilot studies and testing our instrumentation. The screening studies helped us to identify feasible and consistent strategies for sampling candidates and clustering outputs instead of going for arbitrary choices. Encouraging was also that our method revealed a fault in our implementation of the BMI value function (Table 13)[17] Regarding verifying the search procedure and system itself, we use automated unit tests in Julia to mitigate faults in our implementation (https://docs.julialang.org/en/v1/stdlib/Test#Basic-Unit-Tests).

Lastly, our results cannot be generalised beyond the scope of our experimental study, *i.e.,* finding boundaries for unit-level SUTs that take integers as input. On the one hand, various aspects in AutoBVA indicate it might be generally applicable, such as the concept of the program derivative and its basis on string distances rather than type-specific distance function. However, more general test data generation algorithms, mutation operators, and distance functions will need to be experimented with to increase the external validity.

## CONCLUSIONS

While automated boundary value analysis and testing are often advocated or required for quality assurance, prior work has relied mainly on the testers' creativity and experience. As a result, boundary value analysis has been a manual activity. Automated solutions are needed to support testers better, increase efficiency, and make boundary value analysis more systematic. However, existing proposals have relied either on formal specifications, the development of additional models, or white-box analysis, limiting their applicability.

Here we have presented an automated and black-box approach to identify and summarise the boundaries of programs. It is based on coupling a boundary quantification method to a search algorithm. We further proposed using string length distance as a fast but coarse-grained boundary quantifier and proposed two different strategies to search for boundary candidates. Finally, a clustering algorithm was used to summarise the identified candidates in groups based on the similarity of their values.

We validated our approach through two investigations on a total of 613 SUTs with both single and multiple numbers as inputs and arbitrary outputs. We quantitatively evaluated

[17]The implementation did not check for height or weight equals to zero, hence triggering a division by zero error which leads to a NaN output.

how many candidates were found by the two search strategies as well as their uniqueness and diversity. A manual, qualitative analysis of the identified boundary candidates for selecting SUTs was also performed to understand their usefulness better.

We find that even using one of the most straightforward possible boundary quantification metrics, large quantities of boundary candidates could be found by both strategies for a large number of SUTs. While the naive local neighbour search found more unique candidates than the boundary crossing search strategy, the latter found more unique groups if given more search time. Even though our approach is stochastic, it was robust over multiple, repeated executions. The manual analysis showed that many both expected and unexpected boundaries were found. The system identified previously known bugs in the investigated SUTs and the new ones we introduced. Based on our findings, we outlined a simple taxonomy of different actions the proposed system can prompt a tester to take, refining either a test suite, an implementation, or even the specification.

While our results are promising, future work must consider more SUTs that are more complex and have non-numerical input parameters. It should also explore more elaborate search strategies to search globally over the input space and use the already identified candidates and groups to avoid unnecessary, repeated work.

## APPENDIX A. SCREENING STUDY-CONFIGURING THE EXPLORATION STRATEGY

Several parameters greatly influence the exploration process, particularly regarding the sampling strategy. A pre-study investigating two parameters was conducted and is detailed below. The purpose was to reduce the complexity of the main experiments without sacrificing quality, if possible. We here describe the experiments and the verdict.

High-level languages usually offer to sample based on a single concrete datatype (such as Int32 in Julia, representing Integers of 32 bits). When activating CTS, the sampling is based on the compatible types per argument. For abstract data types, the compatible types per argument can be derived from the type graph (see Julia's type graph for numbers in Fig. A1). In this study, all input parameters are Integers. In Julia, Integers are an abstract type that cannot be sampled. Thus we use 128-bit Integers in this study as per default. CTS must contain concrete types only, not abstract types, and a sampler for the type must be available. For instance, for Integer in Julia, the CTS per the graph is

$$CTS(Integer) = \{UInt8, UInt64, UInt32, UInt16, UInt128, Int8, Int64, Int32, Int16,$$
$$Int128, BigInt, Bool\}$$

Further, CTS is not limited to abstract data types such as Integer but can be extended to concrete types. For Int16, and per the conversion rules of Julia, we may declare:

$$CTS(Int16) = \{UInt8, Int8, Int16, Bool\}$$

Sampling using CTS becomes a two-step process. A compatible type is selected for each argument, and then the sampler for that type gets invoked. Two simple approaches to

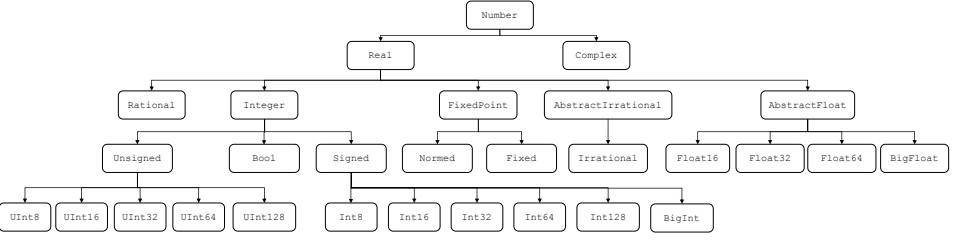

**Figure A1** **The type hierarchy for numbers in Julia shows the compatible types for Integer.** Bool is an Integer.

**Table A1** **The results for the screening over the bytecount SUT.**

| Algorithm | CTS | Sampling | # Found ($\mu \pm \sigma$) |
|---|---|---|---|
| LNS | | uniform | $0.0 \pm 0.0$ |
| LNS | ✓ | uniform | $8.6 \pm 0.9$ |
| LNS | | bituniform | $7.0 \pm 0.0$ |
| LNS | ✓ | bituniform | $9.8 \pm 0.8$ |
| BCS | | uniform | $0.0 \pm 0.0$ |
| BCS | ✓ | uniform | $22.0 \pm 1.0$ |
| BCS | | bituniform | $55.8 \pm 0.8$ |
| BCS | ✓ | bituniform | $56.0 \pm 0.7$ |

choosing the type for each sample are round-robin and uniformly at random. Without loss of generality, we use the latter approach in this study.

We here investigate the impact of CTS (activated, deactivated) and the sampling method (uniform, bituniform), explained in 3.2.1, on the efficacy of boundary exploration. We, therefore, apply AutoBD with LSN compared to BCS according to Section 'Automated Boundary Value Analysis' on bytecount for 30 s each for all possible configurations. As a boundariness measure, we used the program derivative with bytecount on the outputs. The results can be seen in Table A1.

Note that regular uniform sampling *without* CTS, often called random search, rarely finds boundary candidates. First, after activating CTS, boundary candidates can be identified. The greatest number of boundary candidates in all scenarios is obtained when both CTS and bituniform sampling are activated. We, therefore, limit the study to configurations with CTS activated and bituniform sampling. The implied limitations in terms of generalizability are further discussed under 6.

## APPENDIX B. SCREENING STUDY-CLUSTERING FOR SUMMARISATION

The goal of our screening study is to identify a combination of features (U and WD) and distance measures (strlendist, Overlap Coefficient, and Levenshtein) that yield a good model for doing k-means clustering of boundary candidates found by AutoBVA (as

**Table A2  The top 9 clustering configurations for bytecount with their respective average Silhouette score and number of clusters (sorted by Silhouette Score).**

| Configuration | Silhouette score | Number of clusters |
|---|---|---|
| strlendist (WD) + Overlap(WD) | 0.982 | 5 |
| strlendist (WD) + Overlap(WD + U) | **0.942** | **6** |
| strlendist (WD) + Overlap (U) | 0.938 | 3 |
| Overlap(WD + U) | 0.924 | 5 |
| strlendist (WD) + Lev(U) | 0.779 | 3 |
| strlendist (WD) + Overlap(WD + U) + Lev(WD) | 0.777 | 4 |
| Lev(WD) + Overlap(U) | 0.773 | 2 |
| strlendist (WD) + Lev(WD) + Overlap(U) | 0.771 | 3 |
| strlendist (WD) + Overlap(WD) + Lev(U) | 0.760 | 4 |

introduced and explained in 'Setup of summarisation step'). Finding a good clustering is important to allow for the automated and consistent comparison of types of boundary candidates since we cannot define equivalence partitions manually for the chosen SUTs. We also illustrate some examples of features extracted from actual boundary candidates to clarify how the features and attributes represent the different types of boundaries.

To get a clustering of a good fit, we evaluate k in the range of 1–10 and select the one having the highest Silhouette score *Rousseeuw (1987)*, which offers an overview of the cohesion within each cluster and the separation between different clusters *Xiong & Li (2018)*. We use the default level of neighbours for orientation per data point of 15. We use Euclidean distance between feature vectors and 200 max iterations per the interface default. Silhouette values vary between +1 and +1, where +1 indicates that the clusters are clearly distinguishable from each other; values of zero indicate that the clusters are relatively close to each other, so there is little significance in clustering the candidates. Lastly, -1 means that the distance between candidates within the cluster is greater than between different clusters, indicating that the clustering performed is not appropriate and more distinct clusters are likely needed.

We repeated the runs 100 times per all combinations of features, totalling 64, and ranked the configurations according to the silhouette mean while selecting the model of maximum score per configuration. The objective here was not to find perfect-fit models/clusterings but to identify models of a good fit with high discriminatory power because the more clusters that can be differentiated with high accuracy, the more diverse the individual boundary candidates to present to a tester.

The best models can be seen in Table A2. The selected feature set produced the largest number of clusters among the top five percentile Silhouette scores for k-Means over $k \in 1, \ldots, 10$, *i.e.,* bytecount(WD) and bytecount(WD + U). It strikes a good balance between modelling quality and the ability to discriminate for the bytecount SUT that serves for the training due to its low computational complexity and straightforward separation of boundaries within the V domain. The clustering, therefore, uses the Overlap Coefficient for both WD and U. How/whether the feature space generalizes well is a question for future

work. The final matrix with three features represented by four attributes over which all clusterings in this article are conducted is, therefore:

$$M = \begin{bmatrix} |strlendist|(WD) \\ |Overlap|(WD) \\ |Overlap|(U_1) \\ |Overlap|(U_2) \end{bmatrix}.$$

### Funding

The computations were enabled by resources provided by the Swedish National Infrastructure for Computing (SNIC), and by the Swedish Research Council through grant agreement no. 2018-05973. There was no additional external funding received for this study. The funders had no role in study design, data collection and analysis, decision to publish, or preparation of the manuscript.

### Grant Disclosures

The following grant information was disclosed by the authors:
The Swedish National Infrastructure for Computing (SNIC), Swedish Research Council: 2018-05973.

### Competing Interests

The authors declare there are no competing interests.

### Author Contributions

- Felix Dobslaw conceived and designed the experiments, performed the experiments, analyzed the data, performed the computation work, prepared figures and/or tables, authored or reviewed drafts of the article, and approved the final draft.
- Robert Feldt conceived and designed the experiments, analyzed the data, prepared figures and/or tables, authored or reviewed drafts of the article, and approved the final draft.
- Francisco Gomes de Oliveira Neto conceived and designed the experiments, analyzed the data, prepared figures and/or tables, authored or reviewed drafts of the article, and approved the final draft.

### Data Availability

The source code, data and reproduction package is available at Zenodo: Felix Dobslaw. (2023). feldob/AutoBDPaper: reproduction (Version v1). Zenodo. https://doi.org/10.5281/zenodo.7677012

### Supplemental Information

Supplemental information for this article can be found online at http://dx.doi.org/10.7717/peerj-cs.1625#supplemental-information.

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
