# Peer review of "Automated black-box boundary value detection"

_PeerJ Computer Science, doi:10.7717/peerj-cs.1625_

## Round 0.1 · original submission · Minor Revisions

Both reviewers provide several suggestions that should be taken into account in the revision of the paper.

·

Basic reporting

Strengths :
- Overall, the authors propose a novel approach to discover boundary values automatically (to be used as test inputs), based on the concept of a "program derivative" to quantify the "boundariness" of input pairs (from previous work), and a multi-step search algorithm to discover sets of input boundary pairs (this paper).
- Although there is much room for improvement, the experiments conducted showed promising results.
- The paper is generally very well-written and organized, although rather long (but the many details provided may help support further research).


Minor typos :
- l100-101: and (3) -> and
- 107-108: does ... performs poorly -> perform
- 158-160: incomplete phrase?
- 575-578: incomplete phrase?
- 697-700: exporting <-> non-exporting? (to be consistent with Table 5)
- 710: between 5 and 15 -> 6 and 15?
- 808: to a be
- 966: we argue the -> that the
- 976: it is likely .> likely that
- 1318: Integers of base 128 -> 128 bit Integers?
- Table 4: Please solve the "TODO" in the header
- Table 10, Table 11: percentage in percentage -> percentage in parenthesis

Minor clarifications/suggestions:
- Table 4: indicate a single total value per SUT (it is confusing to have pairs of equal totals)
- Table 8: it is not clear how you compute the totals
- Table 10, 11, etc.: please indicate the number of members/pairs in each cluster (and possibly a total)
- 266: can you provide a link to the "bytecount" function source code in Stack Overflow?
- Please describe the "bituniform" sampling technique
- In the abstract, quantitative performance results would be welcome
- The explanation of Algorithm 3 is rather complex (407-429). E.g., one cannot understand why the algorithm tries days 13, 12, 10, 6, -2, and not 13, 12, 11, 10, etc.

Experimental design

Strengths:
- The experimental design is properly described, with well-defined research questions.
- It is provided a replication package with the relevant code and artefacts.

Potential weaknesses/points to discuss:
- Euclidean distance might not be a good choice when you have multiple arguments of very different scales. Please discuss.
- Why don't you compute the mutation score to determine the quality of the results (test data) generated by your AutoBVA method (together with the size of the test suite) and, consequently, provide more evidence to answer RQ3?

Validity of the findings

Strengths:
- The findings and conclusions are properly articulated, with significant detail.

Potential weaknesses/points to discuss:
The following seem to be potential limitations of your approach that are not discussed in the paper:
- Your approach might not be able to test boundary values when there isn't "another side of the boundary". E.g., let's assume you are testing a method that accepts an unsigned integer (being negative values deemed illegal by the compiler). In this case, 0 could be considered a boundary value. But let's assume that the program's output is not significantly different from the next input (1). Then, this boundary might not be discovered by your method. Please discuss.
- Another limitation may result from the fact that you discover the pairs of boundary values based on the actual program behaviour. If the behaviour is faulty in a way that eliminates differences in outputs, then your method might not be able to discover adequate test inputs/boundary values. Please discuss.

Additional comments

Further suggestions:
- To detect candidate pairs, you search the neighbourhood of a given starting point. Couldn't you start with two input points arbitrarily separated, with significantly different program output values, and apply a modified binary search approach to get closer input points while trying to maximize the distance in outputs? In each iteration of the binary search, you would compute the midpoint and keep the left or right value that maximizes the distance in program outputs. The underlying rationale is that it might be easier, in the initial sampling, to hit a high percentage of equivalence classes, rather than a high percentage of points close to boundaries.

Cite this review as
Pascoal Faria J (2023) Peer Review #1 of "Automated black-box boundary value detection (v0.1)". PeerJ Computer Science

Reviewer 2 ·

Basic reporting

The paper is well-written, and the problem is clearly defined. However, sections 3-5 are very dense, with various technical terms, and, therefore, difficult to follow. A very attentive reading is needed to understand what is proposed. I suggest the authors review these sections to make them easier for the reader to comprehend.
Table 4 appears to be incomplete, as indicated in the caption ("TODO update numbers with latest runs"). Additionally, the references cited in lines 1098-1100 need to be adjusted.

Experimental design

no comment

Validity of the findings

The experiments were conducted rigorously and presented in detail.

Additional comments

The authors introduce the AutoBVA approach, aimed at detecting boundary values in the context of black-box testing. The methodology employs a combination of derivative metrics and search and optimization algorithms to identify pairs of possible boundary values automatically. Subsequently, a summarization technique is implemented that considers candidates grouped into VV (where both candidates have valid outputs), VE (where one candidate has a valid output and the other an invalid one), and EE (both candidates have invalid outputs).

The problem's relevance is well-acknowledged, considering the inherent challenge in identifying partitions during functional software testing. One of the approach's strengths lies in its independence from the specification or the source code. However, the research is still in its early stages. It has several limitations, including the restriction of detecting only pairs of candidates that fit into the three mentioned groups and supporting only functions with integer type inputs, limited to three parameters. In its current state, the tool can identify some boundaries but not necessarily all the relevant ones. The distance function for outputs, based on string length, is somewhat simplistic and can present problems when outputs vary but have the same string length.

Nevertheless, the authors are aware of their limitations. Furthermore, the proposed approach can be considered a first step toward automatic partition identification for functional testing.

Cite this review as
Anonymous Reviewer (2023) Peer Review #2 of "Automated black-box boundary value detection (v0.1)". PeerJ Computer Science

---

## Round 0.2 · accepted · Accept

The authors have addressed the issues raised by the reviewers and the paper is now ready for publication.

·

Basic reporting

Strengths :
- Overall, the authors propose a novel approach to discover boundary values automatically (to be used as test inputs), based on the concept of a "program derivative" to quantify the "boundariness" of input pairs (from previous work), and a multi-step search algorithm to discover sets of input boundary pairs (this paper).
- Although there is much room for improvement, the experiments conducted showed promising results.
- The paper is generally very well-written and organized, although rather long (but the many details provided may help support further research).

Weakenesses/points to improve:
In this version, the authors addressed the remarks made in the first review.

Experimental design

Strengths:
- The experimental design is properly described, with well-defined research questions.
- It is provided a replication package with the relevant code and artefacts.

Potential weaknesses/points to discuss:
In this version, the authors discussed the points raised in the first review.

Validity of the findings

Strengths:
- The findings and conclusions are properly articulated, with significant detail.

Potential weaknesses/points to discuss:
In this version, the authors discussed the points raised in the first review.

Additional comments

In the revised version, the authors did a good job of responding satisfactorily to the reviewers' remarks, so I recommend accepting this version.

Cite this review as
Pascoal Faria J (2023) Peer Review #1 of "Automated black-box boundary value detection (v0.2)". PeerJ Computer Science